# FLOW GENERATOR MATCHING

## ABSTRACT

In the realm of Artificial Intelligence Generated Content (AIGC), flow-matching models have emerged as a powerhouse, achieving success due to their robust theoretical underpinnings and solid ability for large-scale generative modeling. These models have demonstrated state-of-the-art performance, but their brilliance comes at a cost. The process of sampling from these models is notoriously demanding on computational resources, as it necessitates the use of multi-step numerical ordinary differential equations (ODEs). Against this backdrop, this paper presents a novel solution with theoretical guarantees in the form of Flow Generator Matching (FGM), an innovative approach designed to accelerate the sampling of flow-matching models into a one-step generation, while maintaining the original performance. On the CIFAR10 unconditional generation benchmark, our one-step FGM model achieves a new record Fréchet Inception Distance (FID) score of 3.08 among few-step flow-matching-based models, outperforming original 50-step flow-matching models. Furthermore, we use the FGM to distill the Stable Diffusion 3, a leading text-to-image flow-matching model based on the MM-DiT architecture. The resulting MM-DiT-FGM one-step text-to-image model demonstrates outstanding industry-level performance. When evaluated on the GenEval benchmark, MM-DiT-FGM has delivered remarkable generating qualities, rivaling other multi-step models in light of the efficiency of a single generation step.

## 1 INTRODUCTIONS

Over the past decade, deep generative models have achieved remarkable advancements across various applications including data generation (Karras et al., 2020b; 2022; Nichol & Dhariwal, 2021; Oord et al., 2016; Ho et al., 2022; Poole et al., 2022; Hoogeboom et al., 2022; Kim et al., 2022), density estimation (Kingma & Dhariwal, 2018; Chen et al., 2019), and image editing (Meng et al., 2021; Couairon et al., 2022). These models have notably excelled in producing high-resolution, text-driven data such as images (Rombach et al., 2022; Saharia et al., 2022; Ramesh et al., 2022; 2021; Luo, 2024), videos (Ho et al., 2022; Brooks et al., 2024), audios (Evans et al., 2024), and others (Zhang et al., 2024; Xue et al., 2023; Luo & Zhang, 2024; Luo et al., 2023b; Zhang et al., 2023; Feng et al., 2023; Deng et al., 2024; Luo et al., 2024c; Geng et al., 2024b; Wang et al., 2024; Pokle et al., 2022), pushing the boundaries of Artificial Intelligence Generated Content (AIGC).

Among the spectrum of deep generative models, flow-matching models (FMs) have emerged as particularly potent, showcasing robust performance in applications like likelihood computation (Grathwohl et al., 2018; Chen et al., 2018) and text-conditional image synthesis(Esser et al., 2024; Liu et al., 2023). Flow models utilize neural networks to parametrize a continuous-time transportation field, establishing a bijective mapping between real data and random prior noises. They are trained to learn conditional vector fields using flow-matching methods (Lipman et al., 2022b; Albergo & Vanden-Eijnden, 2022; Liu et al., 2022; Neklyudov et al., 2023). The flexible parametrization and relative ease of training make FMs versatile across various datasets and applications.

However, despite their strengths, FMs still have severe drawbacks. Primarily, sampling from FMs involves multiple evaluations of the deep neural network, leading to computational inefficiencies. This limitation restricts their broader application, especially in scenarios where efficiency is paramount. Therefore fast sampling from flow models is important though challenging.

Step-wise distillation has emerged as a viable strategy to mitigate the computational inefficiencies associated with iterative sampling processes in deep generative models, particularly for accelerating

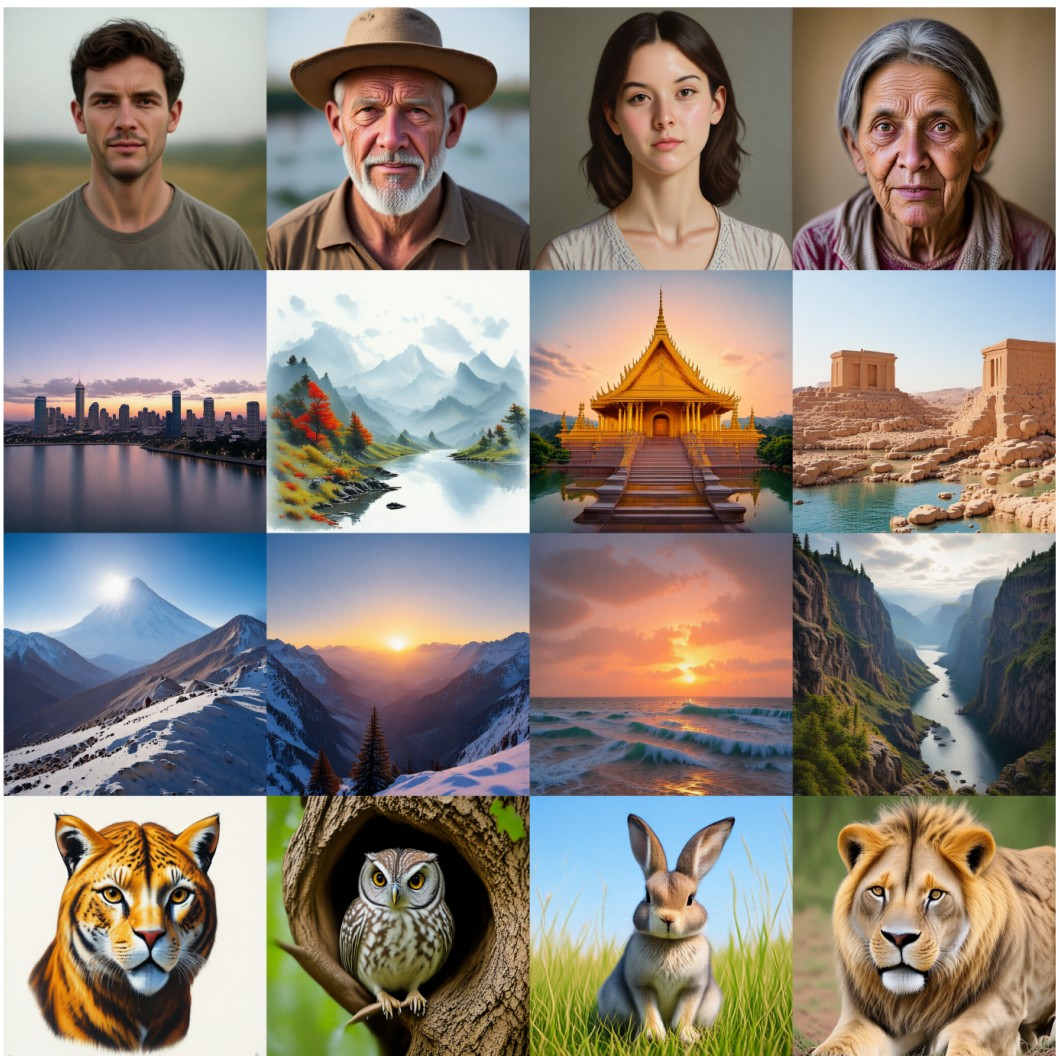

Figure 1: Qualitative Evaluation of one-step samples from MM-DiT-FGM. Prompts used in this figure can be found in the Appendix B.2.1.

diffusion models' sampling mechanisms into more efficient one-step models (Luo et al., 2023a; Salimans & Ho, 2022; Song et al., 2023; Gu et al., 2023a; Fan et al., 2023; Fan & Lee, 2023; Aiello et al., 2023; Watson et al., 2022). While distillation has proven effective in these contexts, the application of such techniques to flow models, has not yet been thoroughly investigated. Besides, since the flow matching does not imply marginal probability densities or score functions as diffusion models do, how to introduce a probabilistic distillation approach for FMs remains challenging.

In this paper, we bridge this gap by presenting *flow generator matching* (FGM), a probabilistic framework for the one-step distillation of flow models. FGM streamlines the sampling process of flow models, making it computationally efficient as a one-step generator, while maintaining high fidelity to the original model's output. Our approach is validated against several benchmarks, such as image generation on the CIFAR10 dataset and large-scale text-to-image generation. On both tasks, we demonstrated very strong performance with only one-step generation. Besides, our experiment on distilling text-to-image flow models shows remarkable performances, marking a new record for one-step text-to-image generation of flow-based models. In conclusion, our exploration not only expands the understanding of distillation techniques but also enhances the practical utility of flow models, particularly in scenarios where quick and efficient sampling is crucial.

## 2 RELATED WORKS

**Diffusion Distillation.** Diffusion distillation (Luo, 2023) is an active research line aiming to accelerate diffusion model sampling using distillation techniques. There are mainly three lines of approaches to distill pre-trained diffusion models to obtain solid few-step models. The first line is the distribution matching method. Luo et al. (2024a) first explore diffusion distillation by minimizing the Integral KL divergence. Yin et al. (2024b) extended this concept by incorporating a data regression loss to enhance performance. Zhou et al. (2024) investigated distillation by focusing on minimizing the Fisher divergence, while Luo et al. (2024b) applied a general score-based divergence to the distillation process. Many other approaches have also studied distribution matching distillation (Nguyen & Tran, 2024; Yuda Song, 2024; Heek et al., 2024; Xie et al., 2024; Xiao et al., 2021; Xu et al., 2024). In this paper, our approach is related to distribution matching distillation. However, how to properly apply distribution matching distillation in the regime of flow models is technically difficult. The second line is the so-called trajectory distillation, which aims to use few-step models to learn the diffusion model's trajectory (Luhman & Luhman, 2021; Salimans & Ho, 2022; Geng et al., 2024a; Meng et al., 2022). Other works use the self-consistency of the diffusion model's trajectory to learn few-step models (Song et al., 2023; Kim et al., 2023; Song & Dhariwal, 2023; Liu et al., 2024; Gu et al., 2023b; Geng et al., 2024b; Salimans et al., 2024).

**Acceleration of Flow Matching Models.** In recent years, there have been efforts to accelerate the sampling process of flow-matching models, most current work focuses on straightening the trajectories of ordinary differential equations (ODEs). ReFlow (Liu et al., 2022) replaces the arbitrary coupling of noise and data originally used for training flow matching with a deterministic coupling generated by a teacher model, enabling the model to learn a rectified flow from the data. CFM (Yang et al., 2024) shares a similar concept with consistency models but differs by applying consistency constraints to the velocity field space instead of the sample space. This approach also serves as a form of regularization aimed at straightening the trajectories of ODEs. Though these works have demonstrated decent accelerations, they are essentially different from our proposed FGM. The FGM is built upon a probabilistic perspective that guarantees the generator distribution matches the teacher FM by minimizing the flow-matching objective. Besides, as we show in Section 5.1, the FGM outperforms the mentioned methods with significant margins.

## 3 BACKGROUNDS

**Flow-matching Models.** Let $\mathbb{R}^d$ represent the data space with data points $\boldsymbol{x} = (\boldsymbol{x}^1, \ldots, \boldsymbol{x}^d) \in \mathbb{R}^d$. Let $q_1(\boldsymbol{x}_1)$ be a simple noise distribution while $q_0(\boldsymbol{x}_0)$ is the data distribution. Let $\boldsymbol{u}_t(\boldsymbol{x}_t|\boldsymbol{x}_0)$ be a known conditional vector field that implies the conditional probabilistic transition $q_t(\boldsymbol{x}_t|\boldsymbol{x}_0)$. The marginal distribution densities $q_t(\boldsymbol{x}_t)$ form a path that links noise distribution $q_1(\boldsymbol{x}_1)$ and data distribution $q_0(\boldsymbol{x}_0)$, i.e. $q_1(\boldsymbol{x}|\boldsymbol{x}_0) = q_1(\boldsymbol{x})$ and $q_0(\boldsymbol{x}|\boldsymbol{x}_0) = \delta(\boldsymbol{x} - \boldsymbol{x}_0)$. Then, one can further define a corresponding marginal vector field (3.2) that translates particles drawn from noise distributions to obtain samples following the data distribution,

$$q_t(\boldsymbol{x}_t) = \int q_t(\boldsymbol{x}_t|\boldsymbol{x}_0)q_0(\boldsymbol{x}_0)\mathrm{d}\boldsymbol{x}_0 \tag{3.1}$$

$$\boldsymbol{u}_t(\boldsymbol{x}_t) = \int \boldsymbol{u}_t(\boldsymbol{x}_t|\boldsymbol{x}_0)\frac{q_t(\boldsymbol{x}_t|\boldsymbol{x}_0)q_0(\boldsymbol{x}_0)}{q_t(\boldsymbol{x}_t)}\mathrm{d}\boldsymbol{x}_0. \tag{3.2}$$

Let $\boldsymbol{v}_\theta(\cdot, \cdot)$ be a vector field parametrized by a deep neural network. The goal of flow matching is to train $\boldsymbol{v}_\theta(\cdot, \cdot)$ to approximate the marginal flow $\boldsymbol{u}_t(\cdot)$ by minimizing the objective (3.3):

$$\mathcal{L}_{FM}(\theta) := \mathbb{E}_{t,\boldsymbol{x}_t \sim q_t(\boldsymbol{x}_t)}\|\boldsymbol{v}_\theta(\boldsymbol{x}_t, t) - \boldsymbol{u}_t(\boldsymbol{x}_t)\|^2. \tag{3.3}$$

Although (3.3) represents the optimal target for optimization, the lack of the explicit expression about $\boldsymbol{u}_t(\boldsymbol{x}_t)$ renders the computation impractical. To address this challenge, Lipman et al. (2022a) introduced flow-matching, a tractable alternative objective of (3.3). Lipman et al. (2022a) shows that one can minimize a simpler yet equivalent objective (3.4):

$$\mathbb{E}_{\substack{t,\boldsymbol{x}_0 \sim q_0(\boldsymbol{x}_0), \\ \boldsymbol{x}_t \sim q_t(\boldsymbol{x}_t|\boldsymbol{x}_0)}} \|\boldsymbol{v}_\theta(\boldsymbol{x}_t, t) - \boldsymbol{u}_t(\boldsymbol{x}_t|\boldsymbol{x}_0)\|^2, \tag{3.4}$$

with $\boldsymbol{x}_t$ is sampled from $q_t(\boldsymbol{x}_t|\boldsymbol{x}_0)$. The main insight of flow-matching is that the tractable objective (3.4) shares the same $\theta$ gradient as (3.3).

**Practical Instance of Flow Matching Models.** In this paper, we especially consider a widely used flow matching model, the rectified flow (ReFlow) (Liu et al., 2022; Albergo & Vanden-Eijnden, 2022) as a specific instance. Our theory and algorithms for the general flow-matching model share the same concepts as the ones based on ReFlow. The ReFlow defines the conditional vector field as

$$\boldsymbol{u}_t(\boldsymbol{x}_t|\boldsymbol{x}_0) = \frac{\boldsymbol{x}_t - \boldsymbol{x}_0}{t} \tag{3.5}$$

This results in a simple training objective as

$$\mathcal{L}_{ReFlow}(\theta) = \mathbb{E}_{\substack{t, \boldsymbol{x}_0 \sim q_0(\boldsymbol{x}_0), \boldsymbol{x}_1 \sim \mathcal{N}(\boldsymbol{0}, \mathbf{I}), \\ \boldsymbol{x}_t = (1-t)\boldsymbol{x}_0 + t\boldsymbol{x}_1}} \|\boldsymbol{v}_\theta(\boldsymbol{x}_t, t) - (\boldsymbol{x}_1 - \boldsymbol{x}_0)\|_2^2 \tag{3.6}$$

The ReFlow objective (3.6) can be interpreted as using a neural network $\boldsymbol{v}_\theta(\boldsymbol{x}_t, t)$ to predict the direction from noises to data samples. In experiment Sections 5.1, we pretrain a flow model in-house using the ReFlow objective (3.6). In Section 5.2, the Stable Diffusion 3 model is also trained with the ReFlow objective.

# 4 FLOW GENERATOR MATCHING

In this section, we introduce Flow Generator Matching (FGM), a general method tailored for the one-step distillation of flow-matching models. We begin by defining problem setup and notations. Then we introduce our matching objective function and how FGM minimizes this objective. Finally, we compare FGM with existing flow distillation approaches, highlighting the empirical and theoretical advantages of our methods.

## 4.1 PROBLEM SETUPS

**Problem Formulation.** Our framework is built upon a pre-trained flow-matching model that accurately approximates the marginal vector field $\boldsymbol{u}_t(\boldsymbol{x}_t)$. The flow $\boldsymbol{u}_t(\boldsymbol{x}_t)$ bridges the noise and data distribution. We also know the conditional transition $q_t(\boldsymbol{x}_t|\boldsymbol{x}_0)$ which implies $\boldsymbol{u}_t(\boldsymbol{x}_t|\boldsymbol{x}_0)$. Assume the pre-trained flow matching model provides a sufficiently good approximation of data distribution, i.e., $q_0$ is the ground truth data distribution.

Our goal is to train a one-step generator model $g_\theta$, which directly transports a random noise $\boldsymbol{z} \sim p_z$ to obtain a sample $\boldsymbol{x}_0 = g_\theta(\boldsymbol{z})$. Let $p_{\theta,0}$ denote the distribution of the student model over the generated sample $\boldsymbol{x}$, and $p_{\theta,t}$ denote the marginal probability path transitioned with $q_t(\boldsymbol{x}_t|\boldsymbol{x}_0)$, i.e.,

$$p_{\theta,t}(\boldsymbol{x}_t) = \int q_t(\boldsymbol{x}_t|\boldsymbol{x}_0)p_{\theta,0}(\boldsymbol{x}_0)d\boldsymbol{x}_0$$

This student marginal probability path implicitly induces a flow vector field $\boldsymbol{v}_{\theta,t}(\boldsymbol{x}_t)$ generating the path, which is unknown yet intractable.

**Intractable Objective.** One-step flow generator matching aims to let the student distribution $p_{\theta,0}$ match the data distribution $q_0$. For this, we consider matching the marginal vector field $\boldsymbol{v}_{\theta,t}$ with the pre-trained one $\boldsymbol{u}_t$ such that the distributions $p_{\theta,0}$ and $q_0$ can match with one another.

In this section, we define the objective for flow generator matching. Based on previous discussions, our goal is to minimize the expected $L^2$ distance between the implicit vector field $\boldsymbol{v}_{\theta,t}$ and the pre-trained flow model's vector field $\boldsymbol{u}_t$, which writes

$$\mathcal{L}_{FM}(\theta) \coloneqq \mathbb{E}_{t, \boldsymbol{x}_t \sim p_{\theta,t}} \|\boldsymbol{v}_{\theta,t}(\boldsymbol{x}_t) - \boldsymbol{u}_t(\boldsymbol{x}_t)\|^2 \tag{4.1}$$

$$= \mathbb{E}_{\substack{t, \boldsymbol{z} \sim p_z(\boldsymbol{z}), \boldsymbol{x}_0 = g_\theta(\boldsymbol{z}), \\ \boldsymbol{x}_t \sim q_t(\boldsymbol{x}_t|\boldsymbol{x}_0)}} \|\boldsymbol{v}_{\theta,t}(\boldsymbol{x}_t) - \boldsymbol{u}_t(\boldsymbol{x}_t)\|^2 \tag{4.2}$$

Notice that the sample $\boldsymbol{x}_t$ is dependent on the parameter $\theta$. We may use $\boldsymbol{x}_t(\theta)$ to emphasize such a parameter reliance if necessary.

It is clear to see that the $\mathcal{L}_{FM}(\theta) = 0$ if and only if all induced vector fields meet, i.e. $\boldsymbol{v}_{\theta,t}(\boldsymbol{x}_t) = \boldsymbol{u}_t(\boldsymbol{x}_t)$ a.s. $p_{\theta,t}$. Therefore it induces that $p_{\theta,t}(\boldsymbol{x}_t) = q_t(\boldsymbol{x}_t)$, a.s. $p_{\theta,t}$, which shows that the two distributions $p_{\theta,0}(\boldsymbol{x}_0) = q_0(\boldsymbol{x}_0)$, a.s. $p_{\theta,0}$ that match with one another. Unfortunately, though minimizing objective (4.1) leads to a one-step generator, it is intractable because we do not know the relation between $\boldsymbol{v}_{\theta,t}(\boldsymbol{x}_t)$ and the generator parameter $\theta$. In the next paragraph, we will bring our main contribution: a tractable yet equivalent training objective as (4.1) with theoretical guarantees.

## 4.2 TRACTABLE OBJECTIVE

Our goal is to optimize the parameter $\theta$ to minimize the objective (4.1). However, the implicit vector field $\boldsymbol{v}_{\theta,t}$ is unknown yet intractable. Therefore it is impossible to directly minimize the objective. However, by taking the gradient of the loss function (4.1) over $\theta$, we have

$$\frac{\partial}{\partial\theta}\mathcal{L}_{FM}(\theta) = \frac{\partial}{\partial\theta}\mathbb{E}_{t,\boldsymbol{x}_t\sim p_{\theta,t}}\|\boldsymbol{u}_t(\boldsymbol{x}_t) - \boldsymbol{v}_{\theta,t}(\boldsymbol{x}_t)\|_2^2$$

$$= \mathbb{E}_{t,\boldsymbol{x}_t\sim p_{\theta,t}}\left\{\frac{\partial}{\partial\boldsymbol{x}_t}\{\|\boldsymbol{u}_t(\boldsymbol{x}_t) - \boldsymbol{v}_{\theta,t}(\boldsymbol{x}_t)\|_2^2\}\frac{\partial\boldsymbol{x}_t(\theta)}{\partial\theta} - 2\{\boldsymbol{u}_t(\boldsymbol{x}_t) - \boldsymbol{v}_{\theta,t}(\boldsymbol{x}_t)\}^T\frac{\partial}{\partial\theta}\boldsymbol{v}_{\theta,t}(\boldsymbol{x}_t)\right\}$$

$$= \mathrm{Grad}_1(\theta) + \mathrm{Grad}_1(\theta). \tag{4.3}$$

Where $\mathrm{Grad}_1(\theta)$ and $\mathrm{Grad}_2(\theta)$ are defined with

$$\mathrm{Grad}_1(\theta) = \mathbb{E}_{t,\boldsymbol{x}_t\sim p_{\theta,t}}\left\{\frac{\partial}{\partial\boldsymbol{x}_t}\{\|\boldsymbol{u}_t(\boldsymbol{x}_t) - \boldsymbol{v}_{\theta,t}(\boldsymbol{x}_t)\|_2^2\}\frac{\partial\boldsymbol{x}_t(\theta)}{\partial\theta}\right\}, \tag{4.4}$$

$$\mathrm{Grad}_2(\theta) = \mathbb{E}_{t,\boldsymbol{x}_t\sim p_{\theta,t}}\left\{-2\{\boldsymbol{u}_t(\boldsymbol{x}_t) - \boldsymbol{v}_{\theta,t}(\boldsymbol{x}_t)\}^T\frac{\partial}{\partial\theta}\boldsymbol{v}_{\theta,t}(\boldsymbol{x}_t)\right\}. \tag{4.5}$$

The gradients in (4.3) consider all derivatives concerning the parameter $\theta$. We put the detailed derivation in Appendix A.1.

Notice that the first gradient $\mathrm{Grad}_1(\theta)$ can be obtained if we stop the $\theta$-gradient for $\boldsymbol{v}_{\theta,t}(\cdot)$, i.e. $\boldsymbol{v}_{\mathrm{sg}[\theta],t}(\cdot)$, This means that we are preventing the gradient of the parameter $\theta$ from propagating through the vector field $\boldsymbol{v}_{\theta,t}$, However, it is important to note that the gradient with respect to $\theta$ can still propagate through $x_t(\theta)$. This results in an alternative loss function whose gradient coincides with $\mathrm{Grad}_1(\theta)$,

$$\mathcal{L}_1(\theta) = \mathbb{E}_{t,\boldsymbol{x}_t\sim p_{\theta,t}}\left\{\|\boldsymbol{u}_t(\boldsymbol{x}_t) - \boldsymbol{v}_{\mathrm{sg}[\theta],t}(\boldsymbol{x}_t)\|_2^2\right\}$$

$$= \mathbb{E}_{\substack{t,\boldsymbol{z}\sim p_z,\boldsymbol{x}_0=g_\theta(\boldsymbol{z}),\\\boldsymbol{x}_t\sim q_t(\boldsymbol{x}_t|\boldsymbol{x}_0)}}\left\{\|\boldsymbol{u}_t(\boldsymbol{x}_t) - \boldsymbol{v}_{\mathrm{sg}[\theta],t}(\boldsymbol{x}_t)\|_2^2\right\} \tag{4.6}$$

However, the second gradient (4.5) involves an intractable term $\frac{\partial}{\partial\theta}\boldsymbol{v}_{\theta,t}(\cdot)$. For the student generator, we only have efficient samples from the conditional probability path, but the vector field $\boldsymbol{v}_{\theta,t}(\cdot)$ along with its $\theta$ gradient is unknown. Fortunately, in this paper we have the following Theorem 4.2, allowing for a more tractable $\theta$-gradient of the student vector field. Before that, we need to first introduce a novel Flow Product Identity in Theorem 4.1, which is one of our contributions.

**Theorem 4.1** (Flow Product Identity). Let $\mathbf{f}(\cdot,\theta)$ be a vector-valued function, using the notations in Section 4.1, under mild conditions, the identity holds:

$$\mathbb{E}_{\boldsymbol{x}_t\sim p_{\theta,t}}\mathbf{f}(\boldsymbol{x}_t,\theta)^T\boldsymbol{v}_{\theta,t}(\boldsymbol{x}_t) = \mathbb{E}_{\substack{\boldsymbol{x}_0\sim p_{\theta,0},\\\boldsymbol{x}_t|\boldsymbol{x}_0\sim q_t(\boldsymbol{x}_t|\boldsymbol{x}_0)}}\mathbf{f}(\boldsymbol{x}_t,\theta)^T\boldsymbol{u}_t(\boldsymbol{x}_t|\boldsymbol{x}_0) \tag{4.7}$$

We put the proof of Flow Product Identity 4.1 in Appendix A.2.

Next, we show that we can introduce an equivalent tractable loss function that has the same parameter gradient as the intractable loss (4.1) in Theorem 4.2.

**Theorem 4.2.** If distribution $p_{\theta,t}$ satisfies some wild regularity conditions, then we have for all $\theta$-parameter free vector-valued function $\boldsymbol{u}_t(\cdot)$, the equation holds for all parameter $\theta$:

$$\mathbb{E}_{\boldsymbol{x}_t\sim p_{\theta,t}}\left\{-2\{\boldsymbol{u}_t(\boldsymbol{x}_t) - \boldsymbol{v}_{\theta,t}(\boldsymbol{x}_t)\}^T\frac{\partial}{\partial\theta}\boldsymbol{v}_{\theta,t}(\boldsymbol{x}_t)\right\}$$

$$= \frac{\partial}{\partial\theta}\mathbb{E}_{\substack{\boldsymbol{x}_0\sim p_{\theta,0},\\\boldsymbol{x}_t|\boldsymbol{x}_0\sim q_t(\boldsymbol{x}_t|\boldsymbol{x}_0)}}\left\{2\{\boldsymbol{u}_t(\boldsymbol{x}_t) - \boldsymbol{v}_{\mathrm{sg}[\theta],t}(\boldsymbol{x}_t)\}^T\{\boldsymbol{v}_{\mathrm{sg}[\theta],t}(\boldsymbol{x}_t) - \boldsymbol{u}_t(\boldsymbol{x}_t|\boldsymbol{x}_0)\}\right\} \tag{4.8}$$

We put the detailed proof in Appendix A.3. The identity (4.8) shows that the expectation of the intractable gradient $\frac{\partial}{\partial\theta}\boldsymbol{v}_{\theta,t}$ can be traded with a tractable expectation with differentiable samples from the student model.

---

**Algorithm 1:** Flow Generator Matching Algorithm for training one-step Generators.

---

**Input:** pre-trained flow matching model $\boldsymbol{u}_t(\cdot)$, one-step generator $g_\theta$, prior distribution $p_z$, online flow model $\boldsymbol{v}_\psi(\cdot)$, time $t \in \mathcal{U}[0,1]$, and conditional transition $q_t(\boldsymbol{x}_t|\boldsymbol{x}_0)$.

**while** *not converge* **do**

freeze $\theta$, update $\psi$ using SGD by minimizing the flow matching loss

$$\mathcal{L}_{FM}(\psi) = \mathbb{E}_{\substack{t, \boldsymbol{z} \sim p_z, \boldsymbol{x}_0 = g_\theta(\boldsymbol{z}), \\ \boldsymbol{x}_t|\boldsymbol{x}_0 \sim q_t(\boldsymbol{x}_t|\boldsymbol{x}_0)}} \|\boldsymbol{v}_\psi(\boldsymbol{x}_t, t) - \boldsymbol{u}_t(\boldsymbol{x}_t|\boldsymbol{x}_0)\|_2^2.$$

freeze $\psi$, update $\theta$ using SGD with by minimizing the FGM loss (4.10):

$$\mathcal{L}_{FGM}(\theta) = \mathcal{L}_1(\theta) + \mathcal{L}_2(\theta)$$

$$\mathcal{L}_1(\theta) = \mathbb{E}_{\substack{t, \boldsymbol{z} \sim p_z, \boldsymbol{x}_0 = g_\theta(\boldsymbol{z}), \\ \boldsymbol{x}_t \sim q_t(\boldsymbol{x}_t|\boldsymbol{x}_0)}} \left\{ \|\boldsymbol{u}_t(\boldsymbol{x}_t) - \boldsymbol{v}_\psi(\boldsymbol{x}_t, t)\|_2^2 \right\} \qquad (4.11)$$

$$\mathcal{L}_2(\theta) = \mathbb{E}_{\substack{t, \boldsymbol{z} \sim p_z, \boldsymbol{x}_0 = g_\theta(\boldsymbol{z}), \\ \boldsymbol{x}_t|\boldsymbol{x}_0 \sim q_t(\boldsymbol{x}_t|\boldsymbol{x}_0)}} \left\{ 2\left\{\boldsymbol{u}_t(\boldsymbol{x}_t) - \boldsymbol{v}_\psi(\boldsymbol{x}_t, t)\right\}^T \left\{\boldsymbol{v}_\psi(\boldsymbol{x}_t, t) - \boldsymbol{u}_t(\boldsymbol{x}_t|\boldsymbol{x}_0)\right\} \right\} \quad (4.12)$$

**end**

**return** $\theta, \psi$.

---

It is a direct result of the identity (4.8) that the gradient $\mathrm{Grad}_2(\theta)$ coincides with the following tractable loss function (4.9) with a stop-graident operation sg imposed on $\theta$ in the generator vector,

$$\mathcal{L}_2(\theta) = \mathbb{E}_{\substack{t, \boldsymbol{z} \sim p_z, \boldsymbol{x}_0 = g_\theta(\boldsymbol{z}), \\ \boldsymbol{x}_t|\boldsymbol{x}_0 \sim q_t(\boldsymbol{x}_t|\boldsymbol{x}_0)}} \left\{ 2\left\{\boldsymbol{u}_t(\boldsymbol{x}_t) - \boldsymbol{v}_{\mathrm{sg}[\theta],t}(\boldsymbol{x}_t)\right\}^T \left\{\boldsymbol{v}_{\mathrm{sg}[\theta],t}(\boldsymbol{x}_t) - \boldsymbol{u}_t(\boldsymbol{x}_t|\boldsymbol{x}_0)\right\} \right\}. \qquad (4.9)$$

Putting together (4.6) and (4.9) in terms of (4.3), we have an equivalent loss to minimize the original objective, that is

$$\mathcal{L}_{FGM}(\theta) = \mathcal{L}_1(\theta) + \mathcal{L}_2(\theta), \qquad (4.10)$$

with $\mathcal{L}_1(\theta)$ and $\mathcal{L}_2(\theta)$ defined in (4.6) and (4.9). This gives rise to the proposed Flow Generator Matching (FGM) objective by minimizing the loss function (4.10). Algorithm 1 summarizes the pseudo algorithm of the flow generator matching by distilling the pre-trained flow matching model into a one-step student generator. It is important to note that the implicit vector field $\boldsymbol{v}_{\theta,t}$ generated by our one-step model still remains intractable. However, since the optimization of $\mathcal{L}_{FGM}(\theta)$ no longer requires the gradient $\frac{\partial}{\partial \theta} \boldsymbol{v}_{\theta,t}(x_t)$, we can effectively utilize an alternative online flow model $\boldsymbol{v}_\psi(\boldsymbol{x}_t, t)$ to take the place of $\boldsymbol{v}_{\mathrm{sg}[\theta],t}(\boldsymbol{x}_t)$, which is inspired by previous works(Luo et al., 2024a; Zhou et al., 2024; Luo et al., 2024b). After our one-step generator $g_\theta$ converged, the online flow model $\boldsymbol{v}_\psi$ is no longer needed.

**Differences From Diffusion Distillations** The FGM gets inspiration from one-step diffusion distillation by minimizing the distribution divergences (Luo et al., 2024a; Zhou et al., 2024; Luo et al., 2024b), however, the resulting theory is essentially different from those of one-step diffusion distillation. The most significant difference between FGM and one-step diffusion distillation is that the flow matching does not imply explicit modeling of either the probability density as the diffusion models do. Therefore, the definitions of distribution divergences can not be applied to flow models as well as its distillation. However, the FGM overcomes such an issue by directly working with the flow-matching objective instead of distribution divergence. The main insight is that our proposed explicit-implicit gradient equivalent theory bypasses the intractable flow-matching objective, resulting in strong practical algorithms with theoretical guarantees. We think Theorem 4.2 may also bring novel contributions to other future studies on flow-matching models.

**Comparison with Other Flow Distillation Methods** There are few existing works that try to accelerate flow models to single-step or few-step generative models. The consistency flow matching (CFM) (Yang et al., 2024) is a most recent work that distills pre-trained flow models into one or two-step models. Though CFM has shown decent results, it is different from our FGM in both

theoretical and practical aspects. First, the theory behind CFM is built on the trajectory consistency of flow models, which is directly generalized from consistency models(Song et al., 2023; Song & Dhariwal, 2023; Geng et al., 2024b). On the contrary, our FGM is motivated by starting from flow-matching objectives, trying to train the one-step generator's implicit flow with the ground truth teacher flow, with theoretical guarantees. On the practical aspects, on CIFAR10 generation, we show that our trained one-step FGM models archive a new SoTA FID of 3.08 among flow-based models, outperforming CFM's best 2-step generation result with an FID of 5.34. Such strong empirical performance marks the FGM as a solid solution for accelerating flow-matching models on standard benchmarks. Besides the toyish CIFAR10 generation, in Section 5.2 we also use FGM to distill leading large-scale text-to-image flow models, obtaining a very robust one-step text-to-image model with almost no performance declines.

## 5 EXPERIMENTS

We conducted experiments to evaluate the effectiveness and flexibility of FGM. Our experiments cover the standard evaluation benchmark, unconditional CIFAR10 image generation, and large-scale text-to-image generation using Stable Diffusion 3 (SD3) (Esser et al., 2024). These experiments demonstrate the FGM's capability to build efficient one-step generators while maintaining high-quality samples.

### 5.1 ONE-STEP CIFAR10 GENERATION

**Experiment Settings.** We first evaluated the effectiveness of FGM on the CIFAR10 dataset (Krizhevsky et al., 2014), the standard testbed for generative model performances. We pre-train flow matching models on CIFAR10 conditional and unconditional generation using ReFlow objective (3.6). We refer to the neural network architecture used for EDM model(Karras et al., 2022). We train both conditional and unconditional models with a batch size of 512 for 20000k images, the resulting in-house-trained flow model shows a CIFAR10 unconditional FID of 2.52 with 300 generation steps, which is slightly worse than the original ReFlow model (Liu et al., 2022) which has an FID of 2.58 using 127 generation steps. However, in Table 1, we find such a slightly worse model does not influence the distillation of a strong one-step generator.

These flow models serve as the teacher models for flow generator matching (FGM). Then we apply FGM to distill one-step generators from flow models. We assess the quality of generated images via Frechet Inception Distance (FID) (Heusel et al., 2017). Lower FID scores indicate higher sample quality and diversity.

Notice that loss (4.11) and loss (4.12) together composite a full parameter gradient of the FGM loss. We find two losses works great for toyish 2D dataset generations using only Multi-layer perceptions. In practice, we find that using loss (4.11) on CIFAR10 models leads to instability, which is a similar observation as Poole et al. (2022) that the condition number of its Jacobian term might be ill-posed. Therefore we do not use loss (4.11) when training and observing good performances. The experiments conducted w and w/o regression loss (4.11) can be found in the Appendix C.2. Training details and hyperparameters are shown in Appendix B.1.

**Initialize Generator with Pretrained Flow Models** Inspired by techniques in diffusion distillation, we initialize the one-step generator with the pre-trained flow models. Recall the flow model's training objective (3.6), the pre-trained flow model $v_\theta(x_t, t)$ approximately predict the direction from random noise to data. Therefore, we use the pre-trained flow to construct our one-step generator. Particularly, we construct the one-step generator with

$$x_0 = (1 - t^*)z + t^* v_\theta(t^* z, t^*), z \sim \mathcal{N}(0, I). \tag{5.1}$$

The $\theta$ is the learnable parameter of the generator, while the $t^*$ is a pre-determined optimal timestep.

**Quantitative Evaluations.** We evaluate each model with the Fretchet Inception Distance (FID) (Heusel et al., 2017), which is a golden standard for evaluating image generation results on the CIFAR10 dataset. Table 1 and Table 2 summarize the FIDs of generative models on CIFAR10 datasets. On unconditional generation, our teacher flow model has an FID of 3.67 and 2.93 with 50 and 100 generation steps respectively. However, our one-step FGM model achieves an FID of **3.08**

Table 1: Unconditional sample quality on CIFAR-10. † means method we reproduced.

| FAMILY | METHOD | NFE ($\downarrow$) | FID ($\downarrow$) |
|---|---|---|---|
| DIFFUSION & GAN | DDPM (HO ET AL., 2020) | 1000 | 3.17 |
| | DD-GAN(T=2) (XIAO ET AL., 2021) | 2 | 4.08 |
| | KD LUHMAN & LUHMAN (2021) | 1 | 9.36 |
| | TDPM (ZHENG ET AL., 2023) | 1 | 8.91 |
| | DFNO (ZHENG ET AL., 2022) | 1 | 4.12 |
| | STYLEGAN2-ADA (KARRAS ET AL., 2020A) | 1 | 2.92 |
| | STYLEGAN2-ADA+DI (LUO ET AL., 2023A) | 1 | 2.71 |
| | EDM (KARRAS ET AL., 2022) | 35 | 1.97 |
| | EDM (KARRAS ET AL., 2022) | 15 | 5.62 |
| | PD (SALIMANS & HO, 2022) | 2 | 5.13 |
| | CD (SONG ET AL., 2023) | 2 | 2.93 |
| | GET (GENG ET AL., 2024A) | 1 | 6.91 |
| | CT (SONG ET AL., 2023) | 1 | 8.70 |
| | iCT-DEEP (SONG & DHARIWAL, 2023) | 2 | 2.24 |
| | DIFF-INSTRUCT (LUO ET AL., 2023A) | 1 | 4.53 |
| | DMD (YIN ET AL., 2024B) | 1 | 3.77 |
| | CTM (KIM ET AL., 2023) | 1 | 1.98 |
| | CTM(KIM ET AL., 2023) | 2 | **1.87** |
| | SiD ($\alpha = 1.0$) (ZHOU ET AL., 2024) | 1 | 1.92 |
| | SiD ($\alpha = 1.2$)(ZHOU ET AL., 2024) | 1 | 2.02 |
| | DI† | 1 | 3.70 |
| FLOW-BASED | 1-REFLOW (+DISTILL) (LIU ET AL., 2022) | 1 | 6.18 |
| | 2-REFLOW (+DISTILL) (LIU ET AL., 2022) | 1 | 4.85 |
| | 3-REFLOW (+DISTILL) (LIU ET AL., 2022) | 1 | 5.21 |
| | CFM(YANG ET AL., 2024) | 2 | 5.34 |
| | **FLOW** | 100 | **2.93** |
| | **FLOW** | 50 | 3.67 |
| | **FGM (OURS)** | 1 | 3.08 |

Table 2: Class-conditional sample quality on CI-FAR10 dataset. † means method we reproduced.

| FAMILY | METHOD | NFE ($\downarrow$) | FID ($\downarrow$) |
|---|---|---|---|
| DIFFUSION & GAN | BIGGAN (BROCK ET AL., 2019) | 1 | 14.73 |
| | BIGGAN+TUNE(BROCK ET AL., 2019) | 1 | 8.47 |
| | STYLEGAN2 (KARRAS ET AL., 2020B) | 1 | 6.96 |
| | MULTIHINGE (KAVALEROV ET AL., 2021) | 1 | 6.40 |
| | FQ-GAN (ZHAO ET AL., 2020) | 1 | 5.59 |
| | STYLEGAN2-ADA (KARRAS ET AL., 2020A) | 1 | 2.42 |
| | STYLEGAN2-ADA+DI (LUO ET AL., 2023A) | 1 | 2.27 |
| | STYLEGAN2 + SMART (XIA ET AL., 2023) | 1 | 2.06 |
| | STYLEGAN-XL (SAUER ET AL., 2022) | 1 | 1.85 |
| | STYLESAN-XL (TAKIDA ET AL., 2023) | 1 | **1.36** |
| | EDM (KARRAS ET AL., 2022) | 35 | 1.82 |
| | EDM (KARRAS ET AL., 2022) | 20 | 2.54 |
| | EDM (KARRAS ET AL., 2022) | 10 | 15.56 |
| | EDM (KARRAS ET AL., 2022) | 1 | 314.81 |
| | GET (GENG ET AL., 2024A) | 1 | 6.25 |
| | DIFF-INSTRUCT (LUO ET AL., 2023A) | 1 | 4.19 |
| | DMD (W.O. REG) (YIN ET AL., 2024B) | 1 | 5.58 |
| | DMD (W.O. KL) (YIN ET AL., 2024B) | 1 | 3.82 |
| | DMD (YIN ET AL., 2024B) | 1 | 2.66 |
| | CTM (KIM ET AL., 2023) | 1 | 1.73 |
| | CTM(KIM ET AL., 2023) | 2 | 1.63 |
| | GDD (ZHENG & YANG, 2024) | 1 | 1.58 |
| | GDD-I (ZHENG & YANG, 2024) | 1 | 1.44 |
| | SiD ($\alpha = 1.0$) (ZHOU ET AL., 2024) | 1 | 1.93 |
| | SiD ($\alpha = 1.2$)(ZHOU ET AL., 2024) | 1 | 1.71 |
| FLOW-BASED | **FLOW** | 100 | 2.87 |
| | **FLOW** | 50 | 3.66 |
| | **FGM (OURS)** | 1 | **2.58** |

using only one generation step, outperforming the teacher model with 50 generation steps with a significant margin of **16%**. On CIFAR10 conditional generation, our one-step FGM model has an FID of **2.58**, outperforming the teacher flow with 100 generation steps which have an FID of **2.87**. In conclusion, our results on CIFAR10 generation benchmarks demonstrate the superior performance of FGM in that it can outperform the multi-step teacher flow model with significant margins.

Besides the strong performances, the training efficiency of FGM is also appealing. In practice, our best one-step FGM model on CIFAR10 unconditional generation is trained with 8 Nvidia A100 GPUs with a batch size of 256. The 1-step FGM reaches an FID of 5.09 (an FID better than converged 2-step CFM) with only 40K images and roughly 7 hours. However, the CFM takes at least 120K images with an even worse FID value of 5.34 with 2 generation steps. On the contrary, the converged FGM shows an FID of 3.08, marking the SoTA among all flow-based few-step models.

The CIFAR-10 generation tasks are much toyish. In Section 5.2, we perform experiments to train large-scale one-step text-to-image generators by distilling from top-performing transformer-based flow models for text-to-image generation. In the next section, we show that the one-step T2I generator distilled by FGM demonstrates state-of-the-art results over other industry-level models.

## 5.2 TEXT-TO-IMAGE GENERATION

**Experiments Settings.** Our goal in this section is to use FGM to train strong one-step text-to-image generators by distillation from leading flow-matching models. For our text-to-image experiments, we selected Stable Diffusion 3 Medium as our teacher model. This model adopts a novel architecture called MMDiT, which enhances performance in image quality, typography, complex prompt understanding, and resource efficiency. For the dataset, we utilized the Aesthetics 6.25+ prompts dataset along with its recaption prompts and sam-recaption data from Chen et al. (2023) for training, comprising approximately 2 million entries. This extensive dataset significantly improves our model's ability to generate high-quality images. Similar to our observation in CIFAR10 generation, we find loss (4.11) leads to unstable training dynamic, therefore we also abandon it when training text-to-image models. For more training details, please refer to Appendix B.2.

**Quantitative Evaluations.** We followed the evaluation metrics used for Stable Diffusion 3 technical report (Esser et al., 2024), and we referenced GenEval metrics to more comprehensively assess the model's response to complex input texts. For the evaluations we conduct, we utilize the configuration recommended by the authors. Our distilled model demonstrates promising results, remaining competitive with other models that require multiple generation steps, even when using only a single generation step.

| SD3 (28 steps) | Hyper-SD3 (4 steps) | Flash-SD3 (4 steps) | Ours (1 step) |

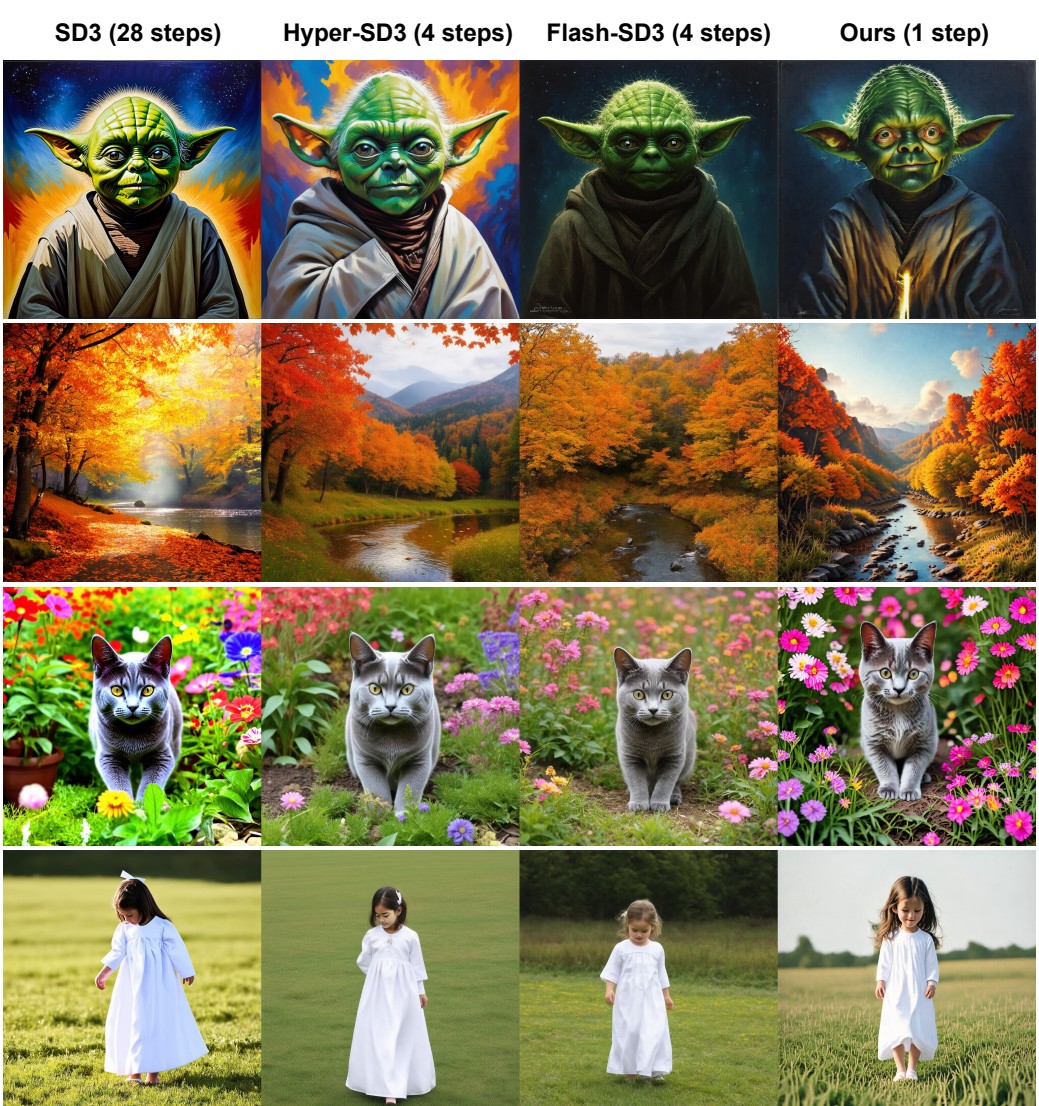

Figure 2: The visual comparison between our **MM-DiT-FGM** and other methods. From left to right, the **first column** is 28-step SD3 model(Esser et al., 2024), the **second column** is the 4-step Hyper-SD3 model(Ren et al., 2024), the **third column** is the 4-step Flash-SD3 model(Chadebec et al., 2024). The prompts for these images are provided in B.2.1

**Qualitative Evaluations.** In this study, we conducted qualitative evaluations of our proposed distillation approach to analyze its performance. Figure 2 showcases several sample outputs, comparing our teacher model, Hyper-SD3(Ren et al., 2024), and Flash-SD3(Chadebec et al., 2024) methods. The results demonstrate high visual quality, particularly in detail and color reproduction, even with only a single generation step. Especially, the one-step MM-DiT-FGM shows aesthetic lightning on each generated image. Compared to existing distillation methods, our model achieves comparable generation quality at a significantly lower cost. Such an advantage makes the FGM plausible in applications when real-time interactions are strictly needed.

**Integration of GAN Loss.** It is clear that the pure FGM algorithm 1 does not rely on any image data when training. In recent years, many studies have shown that incorporating GAN loss into distillation is beneficial for improving high-frequency details on generated images (Yin et al., 2024a; Sauer et al., 2023; 2024). Therefore, we also incorporate a GAN loss with FGM for training one-step text-to-image models and find benefits.

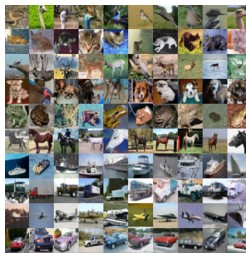 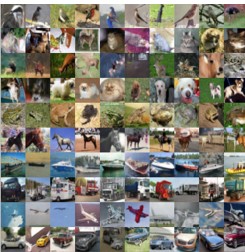 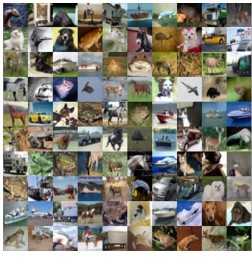 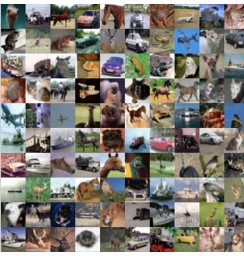

**FGM-1Step CIFAR10 Cond, FID=2.58**  Flow-50Step CIFAR10 Cond, FID=3.66  **FGM-1Step CIFAR10 Uncond, FID=3.08**  Flow-50Step CIFAR10 Uncond, FID=3.67

Figure 3: Visualizations of generated samples from FGM-1step models and 50-step teacher flow models on CIFAR10 datasets. On both conditional and unconditional generation, FGM-1step models outperform 50-step teacher flow models.

| | | Objects | | | | | Color | |
| Model | Overall | Single | Two | Counting | Colors | Position | Attribution | NFEs |
|---|---|---|---|---|---|---|---|---|
| minDALL-E(Zeqiang et al., 2023) | 0.23 | 0.73 | 0.11 | 0.12 | 0.37 | 0.02 | 0.01 | - |
| SD v1.5(Rombach et al., 2022) | 0.43 | 0.97 | 0.38 | 0.35 | 0.76 | 0.04 | 0.06 | 50 |
| PixArt-alpha(Chen et al., 2023) | 0.48 | _0.98_ | 0.50 | 0.44 | 0.80 | 0.08 | 0.07 | 40 |
| SD v2.1(Rombach et al., 2022) | 0.50 | _0.98_ | 0.51 | 0.44 | 0.85 | 0.07 | 0.17 | 50 |
| DALL-E 2 | 0.52 | 0.94 | 0.66 | 0.49 | 0.77 | 0.10 | 0.19 | - |
| SDXL(Podell et al., 2023) | 0.55 | _0.98_ | 0.74 | 0.39 | 0.85 | 0.15 | 0.23 | 50 |
| SDXL Turbo (Sauer et al., 2023) | 0.55 | **1.00** | 0.72 | 0.49 | 0.80 | 0.10 | 0.18 | 1 |
| IF-XL | 0.61 | 0.97 | 0.74 | **0.66** | 0.81 | 0.13 | 0.35 | 100 |
| DALL-E 3(James Betker et al., 2023) | 0.67 | 0.96 | 0.87 | 0.47 | 0.83 | **0.43** | _0.45_ | - |
| SD3†(Esser et al., 2024), | **0.70** | 0.99 | **0.88** | 0.60 | 0.85 | 0.30 | **0.59** | 28 |
| Hyper-SD3†(Ren et al., 2024) | 0.63 | **1.00** | 0.74 | 0.56 | _0.84_ | 0.22 | 0.42 | 4 |
| Flash-SD3†(Chadebec et al., 2024) | 0.67 | 0.99 | 0.77 | _0.59_ | **0.86** | _0.28_ | 0.54 | 4 |
| Ours | _0.65_ | **1.00** | _0.82_ | 0.58 | 0.83 | 0.20 | 0.46 | 1 |

Table 3: **GenEval metrics**. Our distilled model closely matches the performance of the teacher model SD3 (depth=24) on GenEval (Ghosh et al., 2024). Same as Esser et al. (2024) we highlight the **best**, second best, and _third best_ entries. (†indicates that the metrics were evaluated by us.)

During the training process, we observed that in certain intervals of noise schedules where FGM is inefficient, the GAN loss can provide effective gradients to improve the quality of the model's outputs. Therefore, we believe that a significant advantage of GAN loss is its ability to compensate for the inefficiencies of FGM training in certain noise schedules, thereby complementing our loss.

## 6 CONCLUSION

In this paper, we introduce flow-generator matching (FGM), a strong probabilistic one-step distillation approach for flow-matching models. We establish the theoretical foundations of FGM. We also validate the strong empirical performances of FGM on both one-step CIFAR10 generation and large-scale one-step text-to-image generation.

Though FGM has a solid theoretical foundation as well as strong empirical performances, it still has limitations. The first limitation is that currently the FGM still requires an additional flow model that is used for approximating the generator-induced flow vectors. This requirement asks for additional memory costs for distillation and potentially brings challenges when pre-trained flow models and the generators are of large model sizes. Secondly, the FGM is a purely image-data-free approach, which means that it does not need real image data when distilling. However, as a well-known argument, consistently incorporating high-quality image data is important to improve the performances of text-to-image generative models. We hope that future works will explore how to integrate data into the distillation process.

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

## A THEORIES

### A.1 PROOF OF EQUATION 4.3

*Proof.* We prove the equation (4.3) of our loss gradient:

$$\frac{\partial}{\partial\theta}\mathcal{L}_{FM}(\theta) = \frac{\partial}{\partial\theta}\mathbb{E}_{t,\boldsymbol{x}_t\sim p_{\theta,t}}\|\boldsymbol{u}_t(\boldsymbol{x}_t) - \boldsymbol{v}_{\theta,t}(\boldsymbol{x}_t)\|_2^2$$

$$= \mathbb{E}_{t,\boldsymbol{x}_t\sim p_{\theta,t}}\left\{\frac{\partial}{\partial\theta}\|\boldsymbol{u}_t(\boldsymbol{x}_t) - \boldsymbol{v}_{\theta,t}(\boldsymbol{x}_t)\|_2^2\right\}$$

$$= \mathbb{E}_{t,\boldsymbol{x}_t\sim p_{\theta,t}}\left\{2\{\boldsymbol{u}_t(\boldsymbol{x}_t) - \boldsymbol{v}_{\theta,t}(\boldsymbol{x}_t)\}^T\{\frac{\partial\boldsymbol{u}_t(\boldsymbol{x}_t)}{\partial\boldsymbol{x}_t}\cdot\frac{\partial\boldsymbol{x}_t}{\partial\theta} - (\frac{\partial}{\partial\theta}\boldsymbol{v}_{\theta,t}(\boldsymbol{x}_t) + \frac{\partial\boldsymbol{v}_{\theta,t}(\boldsymbol{x}_t)}{\boldsymbol{x}_t}\cdot\frac{\partial\boldsymbol{x}_t}{\partial\theta})\}\right\}$$

$$= \mathbb{E}_{t,\boldsymbol{x}_t\sim p_{\theta,t}}\left\{2\{\boldsymbol{u}_t(\boldsymbol{x}_t) - \boldsymbol{v}_{\theta,t}(\boldsymbol{x}_t)\}^T\{\frac{\partial\boldsymbol{u}_t(\boldsymbol{x}_t)}{\partial\boldsymbol{x}_t}\cdot\frac{\partial\boldsymbol{x}_t}{\partial\theta} - \frac{\partial\boldsymbol{v}_{\theta,t}(\boldsymbol{x}_t)}{\partial\boldsymbol{x}_t}\cdot\frac{\partial\boldsymbol{x}_t}{\partial\theta}\} - 2\{\boldsymbol{u}_t(\boldsymbol{x}_t) - \boldsymbol{v}_{\theta,t}(\boldsymbol{x}_t)\}^T\frac{\partial}{\partial\theta}\boldsymbol{v}_{\theta,t}(\boldsymbol{x}_t)\right\}$$

$$= \mathbb{E}_{t,\boldsymbol{x}_t\sim p_{\theta,t}}\left\{2\{\boldsymbol{u}_t(\boldsymbol{x}_t) - \boldsymbol{v}_{\theta,t}(\boldsymbol{x}_t)\}^T\frac{\partial}{\partial\boldsymbol{x}_t}\{\boldsymbol{u}_t(\boldsymbol{x}_t) - \boldsymbol{v}_{\theta,t}(\boldsymbol{x}_t)\}\cdot\frac{\partial\boldsymbol{x}_t}{\partial\theta} - 2\{\boldsymbol{u}_t(\boldsymbol{x}_t) - \boldsymbol{v}_{\theta,t}(\boldsymbol{x}_t)\}^T\frac{\partial}{\partial\theta}\boldsymbol{v}_{\theta,t}(\boldsymbol{x}_t)\right\}$$

$$= \mathbb{E}_{t,\boldsymbol{x}_t\sim p_{\theta,t}}\left\{\frac{\partial}{\partial\boldsymbol{x}_t}\{\|\boldsymbol{u}_t(\boldsymbol{x}_t) - \boldsymbol{v}_{\theta,t}(\boldsymbol{x}_t)\|_2^2\}\frac{\partial\boldsymbol{x}_t}{\partial\theta} - 2\{\boldsymbol{u}_t(\boldsymbol{x}_t) - \boldsymbol{v}_{\theta,t}(\boldsymbol{x}_t)\}^T\frac{\partial}{\partial\theta}\boldsymbol{v}_{\theta,t}(\boldsymbol{x}_t)\right\}$$

$$(A.1)$$

$\square$

### A.2 PROOF OF THEOREM 4.1

Recall the definition of $p_{\theta,t}$ and $\boldsymbol{v}_{\theta,t}$:

$$p_{\theta,t}(\boldsymbol{x}_t) = \int q_t(\boldsymbol{x}_t|\boldsymbol{x}_0)p_{\theta,0}(\boldsymbol{x}_0)\mathrm{d}\boldsymbol{x}_0 \qquad (A.2)$$

$$\boldsymbol{v}_{\theta,t}(\boldsymbol{x}_t) = \int \boldsymbol{u}_t(\boldsymbol{x}_t|\boldsymbol{x}_0)\frac{q_t(\boldsymbol{x}_t|\boldsymbol{x}_0)p_{\theta,0}(\boldsymbol{x}_0)}{p_{\theta,t}(\boldsymbol{x}_t)}\mathrm{d}\boldsymbol{x}_0. \qquad (A.3)$$

We may use $\mathbf{f}$ for short. We have

$$\mathbb{E}_{\boldsymbol{x}_t\sim p_{\theta,t}}\mathbf{f}^T\boldsymbol{v}_{\theta,t}(\boldsymbol{x}_t) = \mathbb{E}_{\boldsymbol{x}_t\sim p_{\theta,t}}\mathbf{f}^T\int \boldsymbol{u}_t(\boldsymbol{x}_t|\boldsymbol{x}_0)\frac{q_t(\boldsymbol{x}_t|\boldsymbol{x}_0)p_{\theta,0}(\boldsymbol{x}_0)}{p_{\theta,t}(\boldsymbol{x}_t)}\mathrm{d}\boldsymbol{x}_0 \qquad (A.4)$$

$$= \int p_{\theta,t}(\boldsymbol{x}_t)\mathbf{f}^T\int \boldsymbol{u}_t(\boldsymbol{x}_t|\boldsymbol{x}_0)\frac{q_t(\boldsymbol{x}_t|\boldsymbol{x}_0)p_{\theta,0}(\boldsymbol{x}_0)}{p_{\theta,t}(\boldsymbol{x}_t)}\mathrm{d}\boldsymbol{x}_0\mathrm{d}\boldsymbol{x}_t \qquad (A.5)$$

$$= \int\int \mathbf{f}^T\boldsymbol{u}_t(\boldsymbol{x}_t|\boldsymbol{x}_0)q_t(\boldsymbol{x}_t|\boldsymbol{x}_0)p_{\theta,0}(\boldsymbol{x}_0)\mathrm{d}\boldsymbol{x}_0\mathrm{d}\boldsymbol{x}_t \qquad (A.6)$$

$$= \mathbb{E}_{\substack{\boldsymbol{x}_0\sim p_{\theta,0},\\ \boldsymbol{x}_t|\boldsymbol{x}_0\sim q_t(\boldsymbol{x}_t|\boldsymbol{x}_0)}}\mathbf{f}^T\boldsymbol{u}_t(\boldsymbol{x}_t|\boldsymbol{x}_0) \qquad (A.7)$$

### A.3 PROOF OF THEOREM 4.2

*Proof.* Let us take $\theta$ gradient on both sides of (4.7), and then we have

$$\mathbb{E}_{\boldsymbol{x}_t\sim p_{\theta,t}}\left\{\frac{\partial}{\partial\theta}\mathbf{f}(\boldsymbol{x}_t,\theta)^T\boldsymbol{v}_{\theta,t}(\boldsymbol{x}_t) + \mathbf{f}(\boldsymbol{x}_t,\theta)^T\frac{\partial}{\partial\theta}\boldsymbol{v}_{\theta,t}(\boldsymbol{x}_t)\right\} + \mathbb{E}_{\boldsymbol{x}_t\sim p_{\theta,t}}\frac{\partial}{\partial\boldsymbol{x}_t}\left\{\mathbf{f}(\boldsymbol{x}_t,\theta)^T\boldsymbol{v}_{\theta,t}(\boldsymbol{x}_t)\right\}\frac{\partial\boldsymbol{x}_t}{\partial\theta}$$

$$(A.8)$$

$$= \mathbb{E}_{\substack{\boldsymbol{x}_0\sim p_{\theta,0},\\ \boldsymbol{x}_t|\boldsymbol{x}_0\sim q_t(\boldsymbol{x}_t|\boldsymbol{x}_0)}}\frac{\partial}{\partial\theta}\mathbf{f}(\boldsymbol{x}_t,\theta)^T\boldsymbol{u}_t(\boldsymbol{x}_t|\boldsymbol{x}_0) + \mathbb{E}_{\substack{\boldsymbol{x}_0\sim p_{\theta,0},\\ \boldsymbol{x}_t|\boldsymbol{x}_0\sim q_t(\boldsymbol{x}_t|\boldsymbol{x}_0)}}\left\{\frac{\partial}{\partial\boldsymbol{x}_t}\left[\mathbf{f}(\boldsymbol{x}_t,\theta)^T\boldsymbol{u}_t(\boldsymbol{x}_t|\boldsymbol{x}_0)\right]\frac{\partial\boldsymbol{x}_t}{\partial\theta}\right.$$

$$\left. + \mathbf{f}(\boldsymbol{x}_t,\theta)^T\frac{\partial}{\partial\boldsymbol{x}_0}\boldsymbol{u}_t(\boldsymbol{x}_t|\boldsymbol{x}_0)\frac{\partial\boldsymbol{x}_0}{\partial\theta}\right\}$$

Notice that one can have

$$\mathbb{E}_{\boldsymbol{x}_t\sim p_{\theta,t}}\left\{\frac{\partial}{\partial\theta}\mathbf{f}(\boldsymbol{x}_t,\theta)^T\right\}\boldsymbol{v}_{\theta,t}(\boldsymbol{x}_t) = \mathbb{E}_{\substack{\boldsymbol{x}_0\sim p_{\theta,0},\\ \boldsymbol{x}_t|\boldsymbol{x}_0\sim q_t(\boldsymbol{x}_t|\boldsymbol{x}_0)}}\frac{\partial}{\partial\theta}\mathbf{f}(\boldsymbol{x}_t,\theta)^T\boldsymbol{u}_t(\boldsymbol{x}_t|\boldsymbol{x}_0)$$

by substituting $\mathbf{f}(\boldsymbol{x}_t, \theta)$ with $\frac{\partial}{\partial\theta}\mathbf{f}(\boldsymbol{x}_t, \theta)$ in equation (4.7).

This allows us to cancel out the corresponding terms from equation (A.8), and we have

$$\mathbb{E}_{\boldsymbol{x}_t \sim p_{\theta,t}}\left\{\mathbf{f}(\boldsymbol{x}_t, \theta)^T \frac{\partial}{\partial\theta}\boldsymbol{v}_{\theta,t}(\boldsymbol{x}_t)\right\} + \mathbb{E}_{\boldsymbol{x}_t \sim p_{\theta,t}}\frac{\partial}{\partial\boldsymbol{x}_t}\left\{\mathbf{f}(\boldsymbol{x}_t, \theta)^T \boldsymbol{v}_{\theta,t}(\boldsymbol{x}_t)\right\}\frac{\partial\boldsymbol{x}_t}{\partial\theta} \tag{A.9}$$

$$= \mathbb{E}_{\substack{\boldsymbol{x}_0 \sim p_{\theta,0}, \\ \boldsymbol{x}_t|\boldsymbol{x}_0 \sim q_t(\boldsymbol{x}_t|\boldsymbol{x}_0)}}\left\{\frac{\partial}{\partial\boldsymbol{x}_t}\left[\mathbf{f}(\boldsymbol{x}_t, \theta)^T \boldsymbol{u}_t(\boldsymbol{x}_t|\boldsymbol{x}_0)\right]\frac{\partial\boldsymbol{x}_t}{\partial\theta} + \mathbf{f}(\boldsymbol{x}_t, \theta)^T\frac{\partial}{\partial\boldsymbol{x}_0}\boldsymbol{u}_t(\boldsymbol{x}_t|\boldsymbol{x}_0)\frac{\partial\boldsymbol{x}_0}{\partial\theta}\right\}$$

This gives rise to

$$\mathbb{E}_{\boldsymbol{x}_t \sim p_{\theta,t}}\left\{\mathbf{f}(\boldsymbol{x}_t, \theta)^T \frac{\partial}{\partial\theta}\boldsymbol{v}_{\theta,t}(\boldsymbol{x}_t)\right\} \tag{A.10}$$

$$= \mathbb{E}_{\substack{\boldsymbol{x}_0 \sim p_{\theta,0}, \\ \boldsymbol{x}_t|\boldsymbol{x}_0 \sim q_t(\boldsymbol{x}_t|\boldsymbol{x}_0)}}\left\{\frac{\partial}{\partial\boldsymbol{x}_t}\left[\mathbf{f}(\boldsymbol{x}_t, \theta)^T\left\{\boldsymbol{u}_t(\boldsymbol{x}_t|\boldsymbol{x}_0) - \boldsymbol{v}_\theta(\boldsymbol{x}_t, t)\right\}\right]\frac{\partial\boldsymbol{x}_t}{\partial\theta} + \mathbf{f}(\boldsymbol{x}_t, \theta)^T\frac{\partial\boldsymbol{u}_t(\boldsymbol{x}_t|\boldsymbol{x}_0)}{\partial\boldsymbol{x}_0}\frac{\partial\boldsymbol{x}_0}{\partial\theta}\right\}$$

We now define the following loss function

$$\mathcal{L}_2(\theta) = \mathbb{E}_{\substack{\boldsymbol{x}_0 \sim p_{\theta,0}, \\ \boldsymbol{x}_t|\boldsymbol{x}_0 \sim q_t(\boldsymbol{x}_t|\boldsymbol{x}_0)}}\left\{\mathbf{f}(\boldsymbol{x}_t, \mathrm{sg}[\theta])^T\left\{\boldsymbol{u}_t(\boldsymbol{x}_t|\boldsymbol{x}_0) - \boldsymbol{v}_{\mathrm{sg}[\theta],t}(\boldsymbol{x}_t)\right\}\right\} \tag{A.11}$$

with $\mathbf{f}(\boldsymbol{x}_t, \theta) = -2\left\{\boldsymbol{u}_t(\boldsymbol{x}_t) - \boldsymbol{v}_{\theta,t}(\boldsymbol{x}_t)\right\}$. Its gradient becomes

$$\mathbb{E}_{\boldsymbol{x}_t \sim p_{\theta,t}}\left\{-2\left\{\boldsymbol{u}_t(\boldsymbol{x}_t) - \boldsymbol{v}_{\theta,t}(\boldsymbol{x}_t)\right\}^T\frac{\partial}{\partial\theta}\boldsymbol{v}_{\theta,t}(\boldsymbol{x}_t)\right\}$$

$$= \frac{\partial}{\partial\theta}\mathbb{E}_{\substack{\boldsymbol{x}_0 \sim p_{\theta,0}, \\ \boldsymbol{x}_t|\boldsymbol{x}_0 \sim q_t(\boldsymbol{x}_t|\boldsymbol{x}_0)}}\left\{\mathbf{f}(\boldsymbol{x}_t, \mathrm{sg}[\theta])^T\left\{\boldsymbol{u}_t(\boldsymbol{x}_t|\boldsymbol{x}_0) - \boldsymbol{v}_{\mathrm{sg}[\theta]}(\boldsymbol{x}_t, t)\right\}\right\} \tag{A.12}$$

by applying the above result in (A.10). This completes the proof of Theorem 4.1, and shows the gradient of $\mathcal{L}_2(\theta)$ coincides with $\mathrm{Grad}_2(\theta)$.

$\square$

# B  ADDITIONAL EXPERIMENTAL DETAILS

## B.1  CIFAR-10

**Hyper-parameters**  Please note that prior to distilling our one-step flow matching models, we first pre-trained multi-step flow matching models on CIFAR-10 using the ReFlow objective. All experimental details can be found in Table 4.

When distilling our one-step model, we use a logit-normal distribution $\pi(0, 2)$. Larger variance allows the training to cover a wider range of noise levels, which provides better stability for the training process. Excessively high noise level can lead to a decline in the quality of the generated images, while excessively low noise can easily result in mode collapse issues.

Table 4: Experimental details on CIFAR-10.

| Training Details | CIFAR-10 Uncond | CIFAR-10 Cond | CIFAR-10 Uncond (1 Step) | CIFAR-10 Cond (1 Step) |
|---|---|---|---|---|
| Training Kimg | 20000 | 20000 | 20000 | 20000 |
| Batch size | 512 | 512 | 512 | 512 |
| Optimizer ($\boldsymbol{v}_\psi$) | Adam | Adam | Adam | Adam |
| Optimizer ($g_\theta$) | Adam | Adam | Adam | Adam |
| Learning rate ($\boldsymbol{v}_\psi$) | 2e-5 | 2e-5 | 2e-5 | 2e-5 |
| Learning rate ($g_\theta$) | 2e-5 | 2e-5 | 2e-5 | 2e-5 |
| betas ($\boldsymbol{v}_\psi$) | (0, 0.999) | (0, 0.999) | (0, 0.999) | (0, 0.999) |
| betas ($g_\theta$) | (0, 0.999) | (0, 0.999) | (0, 0.999) | (0, 0.999) |
| EMA decay rate | 0.999 | 0.999 | 0.999 | 0.999 |

## B.2 TEXT-TO-IMAGE

**Hyper-parameters** We detail the hyperparameters used in the distillation of our text-to-image models, specifically for both the one-step generator and the online flow model. Both models are trained in BF16 precision using the Adam optimizer with the following settings: $\beta_1 = 0, \beta_2 = 0.999, \epsilon = 1.0 \times 10^{-6}$, and a learning rate of $5.0 \times 10^{-6}$. For both the FGM loss and the flow matching loss, we sample timestep $t \in [0, 1]$, following the Esser et al. (2024) using a logit-normal distribution as the timestep density function. The FGM loss employs $\pi(2.4, 1.0)$, while the flow matching loss uses $\pi(-1.0, 2.0)$. During the generator training phase, the GAN loss weight is set to $1 \times 10^{-2}$, whereas for the discriminator training, it is set to $5 \times 10^{-2}$. Additionally, we apply a loss scaling factor of 100 for the generator, and the entire model is trained with a batch size of 192.

**Training Details** During the training of the generator, we employed classifier-free guidance for inference on the teacher model when calculating $\mathcal{L}_2(\theta)$. To prevent artifacts in the output caused by an excessively high guidance scale, we opted for a more stable guidance scale of 4.0. To further reduce memory consumption, we pre-encoded the prompts dataset into embeddings. For the negative prompts used in classifier-free guidance, we used empty text for encoding and storage. Additionally, by applying Fully Sharded Data Parallel (FSDP) across the teacher model, online flow model, and generator, we achieved a batch size of 4 with a gradient accumulation of 6, ultimately allowing us to reach a batch size of 192 on 8xH800-80G.

**Discriminator Design** For the design of the discriminator's network architecture, we drew on previous work (Yin et al., 2024a), using the online flow model itself as a feature extractor for images, supplemented by a lightweight convolutional network as the classification head to differentiate between the distributions of noisy real data and generated data. However, unlike Yin et al. (2024a), the teacher model we chose does not have an explicit encoder structure. As a result, we output the hidden states from different layers of the transformer and found that the shallow features, specifically those from layer 2, better reflect the content of the image compared to deeper layers. Thus, we empirically selected this layer's features as the input for the subsequent classification head. Additionally, as mentioned earlier, we discovered that GAN can perform well in certain noise ranges where FGM is inefficient. Therefore, another distinction from Yin et al. (2024a) is our different design for the noise schedules used for FGM and GAN loss. The former primarily samples in high-noise ranges, while the latter focuses on sampling in lower-noise ranges. GAN training is conducted on a synthetic dataset containing approximately 500K high-quality images at a resolution of 1024px. Texts and images have also been pre-encoded and stored to reduce computational load during training.

**Model Parameterization** The standard flow-matching model for generating data from noise can be represented in EDM formulation as follows:

$$\boldsymbol{x}_0 = c_{\text{skip}} \cdot \boldsymbol{z} - c_{\text{out}} \cdot \boldsymbol{v}_\theta(c_{\text{in}} \cdot \boldsymbol{z}, t), \quad \boldsymbol{z} \sim \mathcal{N}(\boldsymbol{0}, \mathbf{I}) \tag{B.1}$$

Generally, the conventional choices for one-step generator are $t = t^* = 1, c_{\text{skip}} = 1, c_{\text{out}} = t^*, c_{\text{in}} = 1$. However, in practice, we identified two empirical modifications to these parameters that can further enhance the model's generation performance.

First, regarding the choice of $t^*$, since we need to inherit weights from the teacher model, selecting $t^*$ effectively means choosing a specific $v_{\theta, t^*}$ from a family of models with shared parameters $v_{\theta, t}$. To optimize our initialization weights, we can select the model that performs best for one-step generation within this family. Given a simple hyperparameter search, we noticed that $t^* = 0.97$ is a good choice.

Second, we examined the input scaling factor $c_{\text{in}}$. While the standard choice is $c_{\text{in}} = 1$, we noticed during our training that the generated results consistently contained some small noise and blurriness that were difficult to eliminate. After multiple tuning attempts, we suspected that the variance of the model input was too large. We decided to slightly reduce the input variance and chose $c_{\text{in}} = 0.8$. Consequently, we derived our final model parameterization:

$$\boldsymbol{x}_0 = \boldsymbol{z} - 0.97 \cdot \boldsymbol{v}_\theta(0.8 \cdot \boldsymbol{z}, 0.97)), \quad \boldsymbol{z} \sim \mathcal{N}(\boldsymbol{0}, \mathbf{I}) \tag{B.2}$$

**Our Evaluation Settings** In our evaluation, we evaluated several other models on GenEval(Ghosh et al., 2024), including SD3-Medium(Esser et al., 2024), Hyper-SD3(Ren et al., 2024), and Flash-SD3Chadebec et al. (2024). All evaluations were conducted at a resolution of 1024px, generating

four samples for each prompt from the original GenEval paper. We utilized the inference parameters recommended by the authors for these models. Specifically, for SD3, we use a guidance scale of 7.0, generating images in 28 steps.For Hyper-SD3, we applied a guidance scale of 3.0 and a LoRA scale of 0.125, performing 4 steps of inference to generate the evaluation images. For Flash-SD3, we set the guidance scale to 0.0 and also used 4 sampling steps. Finally, we automatically calculated the corresponding metrics using the scripts provided by GenEval.

### B.2.1 EVALUATION PROMPTS

**Prompts used in Figure 1**

- *blurred landscape, close-up photo of man, 1800s, dressed in t-shirt.*
- *Seasoned fisherman portrait, weathered skin etched with deep wrinkles, white beard, piercing gaze beneath a fisherman's hat, softly blurred dock background accentuating rugged features, captured under natural light, ultra-realistic, high dynamic range photo.*
- *Portrait of a Young Woman.*
- *an old woman, Eyes Wide Open, Siena International Photo Awards.*
- *View of Perth City skyline at dusk.*
- *Chinese landschap aquarel.*
- *Wood Print featuring the photograph Gold Temple, by Rikk Flohr*
- *The Ruins at Philae Egypt*
- *Arequipa and an Ascent of Volcan Chachani, Highlux Photography*
- *This was one of the most striking alpine sunrises that I have witnessed and despite cold and wind...*
- *Lets stay a while longer, rough ocean at sunset*
- *Gorge Light - Oregon*
- *Airbrushed Animals by Eyan Higgins Jones*
- *Staande foto Uil Bird, Owl, Three Spotted owlet (Athene brama) in tree hollow, Bird of Thailand*
- *A fluffy rabbit sitting upright in a field of tall grass, ears perked up and alert, with a bright blue sky above.*
- *The lion was shot dead after the person was killed.*

**Prompts used in Figure 2**

- *Luminous Beings Are We painting by Stephen Andrade Gallery1988 Star Wars Art Awakens Yoda*
- *Delightful Fall Landscape Wallpapers*
- *Russian Blue cat exploring a garden, surrounded by vibrant flowers.*
- *A young girl walks across a field, head down, wearing a communion gown.*

### B.2.2 MORE SAMPLES

## C ABLATION STUDY

### C.1 GENERATOR INITIALIZATION

In our practical experience, we have discovered that the initialization of the generator has a substantial impact on the convergence of model training. Previous studies(Luo et al., 2024a; Chen et al., 2024; Zhou et al., 2024; Yin et al., 2024a) on diffusion models indicated that the initialization of $t^*$ should be situated near the beginning and middle segments of the scheduler. In contrast, our experiments with flow-matching reveal that the most suitable range for $t^*$ is located towards the latter

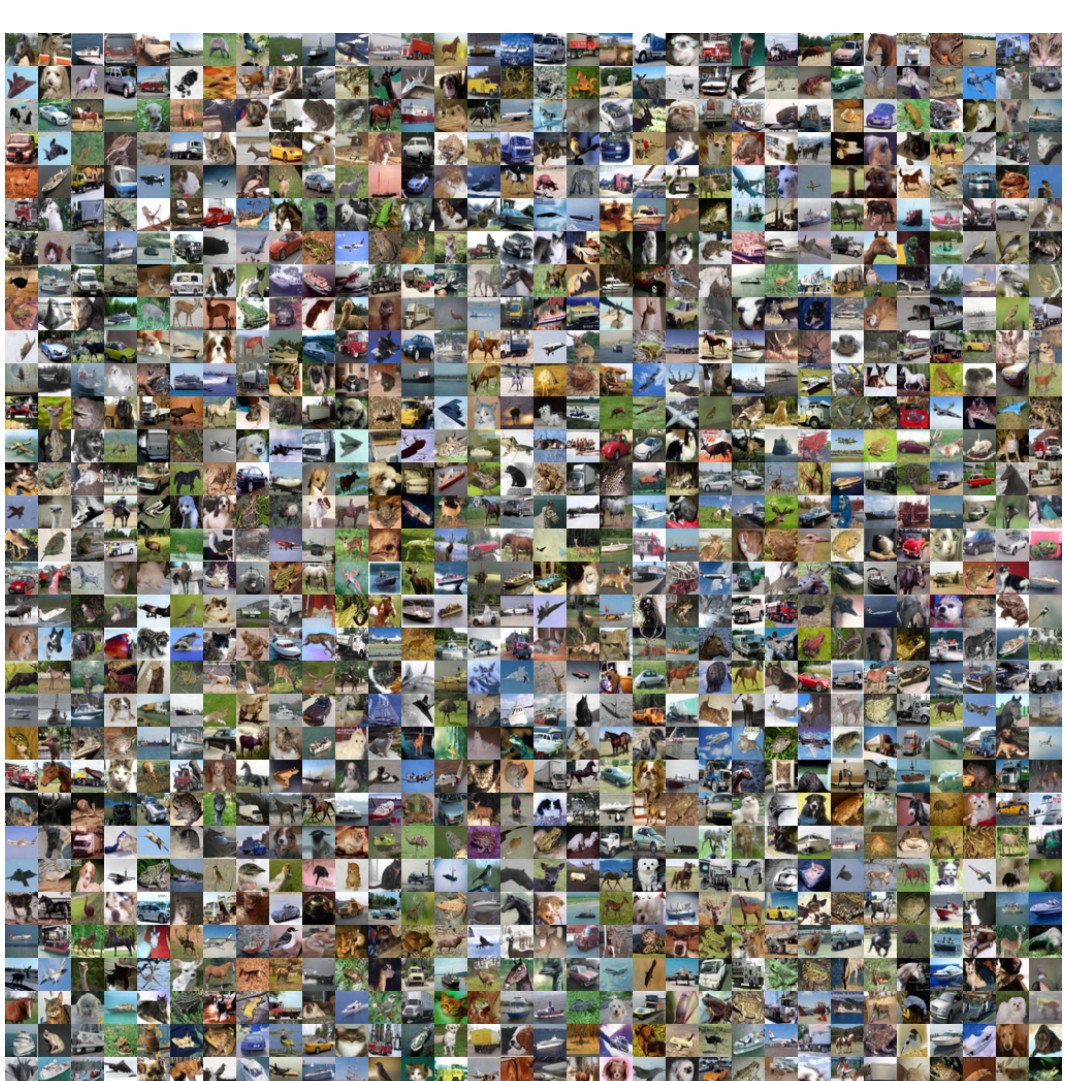

Figure 4: Unconditional samplers from 1-step FGM model on CIFAR10.

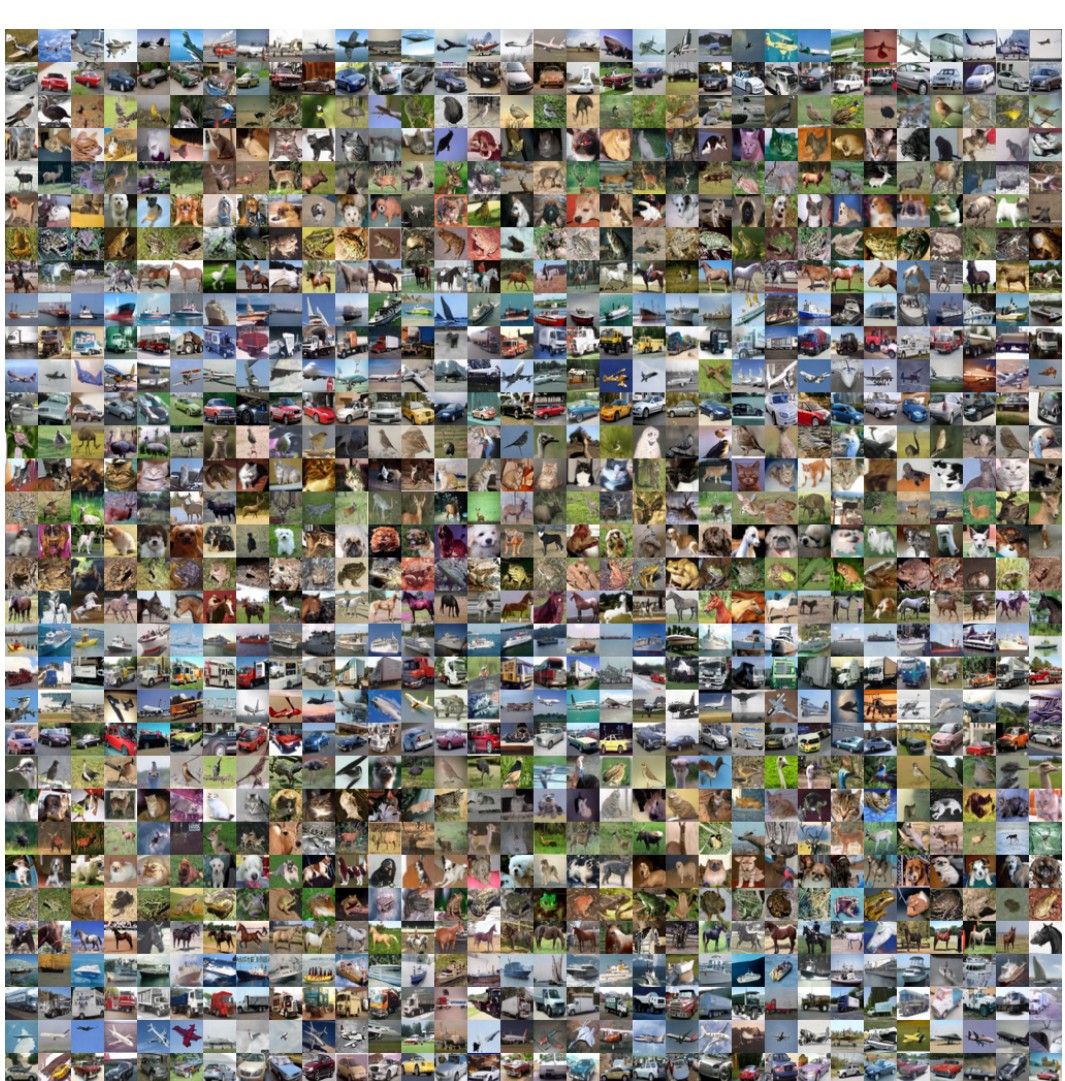

Figure 5: Conditional samplers from 1-step FGM model on CIFAR10.

part of the process. In our experiments, we choose several $t^* = [0.00, 0.25, 0.50, 0.75, 1.00]$ to train from scratch on 512-px, and the qualitative results are presented in Fig 6. Notes that our model parameterization for the ablation can be simplified as

$$\widehat{\boldsymbol{x}}_0 = \boldsymbol{z} - \boldsymbol{v}_\theta(\boldsymbol{z}, t^*), \quad \boldsymbol{z} \sim \mathcal{N}(\boldsymbol{0}, \mathbf{I}) \tag{C.1}$$

The visual results indicate that a suitable range for $t^*$ should be $[0.75, 1.00]$. However, the cost of further determining the optimal choice for $t^*$ is likely to be high and may not yield significant value. A key observation is that as $t^*$ decreases, the structural integrity of the images tends to deteriorate. This phenomenon can be attributed to the property of pre-trained flow matching model. When noise intensity is high, the model primarily focuses on generating the overarching structure of the image. Conversely, at lower noise intensity, the model leans toward creating finer details based on the pre-existing structure. However, in our one-step model, this foundational structure is absent, resulting in divergence.

## C.2 TRAINING WITH REGRESSION LOSS

In our training, we excluded regression loss $\mathcal{L}_1$ based on experience. To further illustrate its impact on the training process, we conduct two experiments on an early checkpoints, one training with both loss $\mathcal{L}_1 + \mathcal{L}_2$, another training with only $\mathcal{L}_2$, our results in Fig 7 show that simply apply the extra regression loss $\mathcal{L}_1$ quickly degrade the performance. From the visual results we can tell that the model trained with $\mathcal{L}_1$ resulting noisy images and quickly corrupted. So the regression term is omitted in our training.

## D IMAGE QUALITY IMPROVEMENT BY FURTHER TRAINING

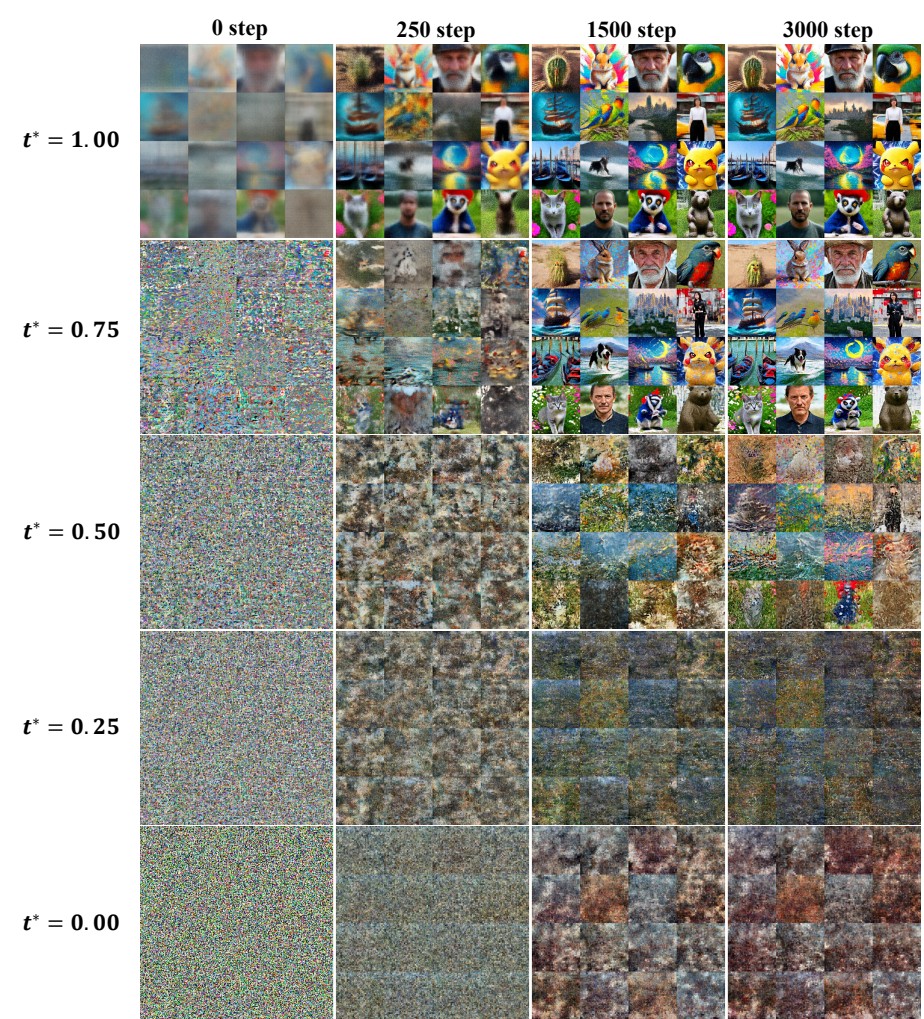

Figure 6: We choose several $t^* = [0.00, 0.25, 0.50, 0.75, 1.00]$ to train from scratch on 512-px. As $t^*$ decreases, the structural integrity of the images tends to deteriorate.

**After 1 Step Optimization**    **After 250 Step Optimization**

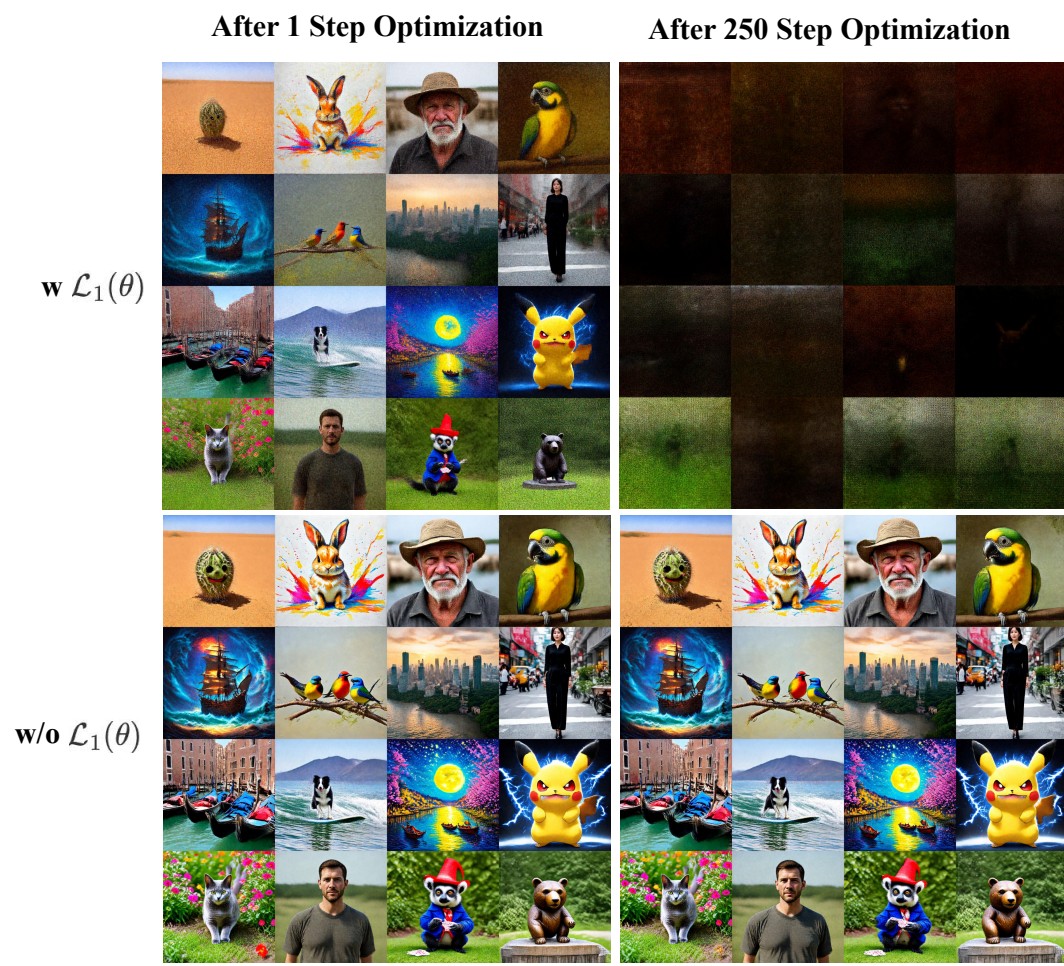

Figure 7: We conduct two experiments on an early checkpoints, one training with both loss $\mathcal{L}_1 + \mathcal{L}_2$, another training with only $\mathcal{L}_2$, our results show that simply apply the extra regression loss $\mathcal{L}_1$ quickly degrade the performance.

**Previous**     **More Steps**     **Previous**     **More Steps**

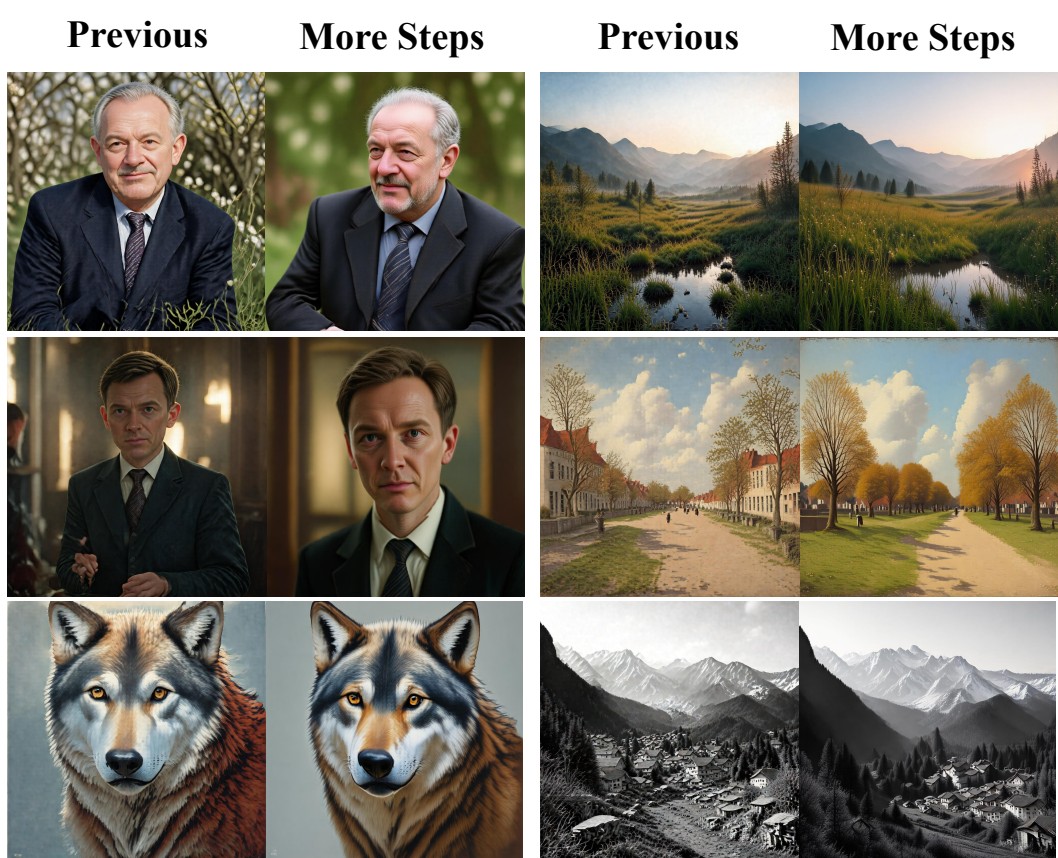

Figure 8: This suggests the checkerboard artifacts can be substantially mitigated, and the overall image quality can also be enhanced with more extensive training.

