# OpenReview forum: "One-step Flow Matching Generators"
_ICLR.cc/2025/Conference — Submitted to ICLR 2025_

### Official Review · Reviewer_BVRU · 2024-10-30

**Soundness:** 2
**Presentation:** 1
**Contribution:** 3
**Rating:** 5
**Confidence:** 3

**Summary:**

The paper introduces Flow Generator Matching (FGM), a method for to distill a pretrained flow-matching model into a single-step model. Specifically, the paper proposes a data-free distillation approach to match the implicit vector field of the student model $v_{\theta, t}$ with the pretrained vector field u_t, where u_t can be regarded as the teacher. Experiments are carried out on CIFAR-10 and paired text-image dataset, which demonstrate the effectiveness.

**Strengths:**

1. The paper proposes a novel method to distill a flow-matching model based on the property of its vector field from a probabilistic standpoint, which represents a good contribution.
2. Experiments show effectiveness of the appraoch.

**Weaknesses:**

1. My biggest concern is the clarity of the paper.
    - Algorithm 1 is not clear to me and hard to understand. What is the online flow model $v_\psi$? It gets introduced nowhere in the entire Section 4 except the Input field of Algorithm 1. Does it represent the implicit vector field? If yes, why shall we have another notation of $v_{sg[\theta], t}$ in 4.11 and 4.12? If not, then what does the online model do and how could we use $\theta$ to parameterize both the generator and the implicit vector field? Also in 4.12, there're both $u_t(x_t)$ and $u_t(x_t | x_0)$, yet $x_t$ is only sampled from $x_t | x_0$ ~ $q_t(x_t | x_0)$, then why do we have different notations here? This is the core of the paper which concretizes the theory into algorithm, and it shall be presented as clear as possible.
    - in line 203, there's an abuse of notation where using $x_0 = g_\theta(z)$ instead of $x$ could make the notation more consistent with the latter integral.
2. The one-step results seem to have checkerboard artifact. When zoomed in Fig.1, most samples show the pattern. It also appear in the 2nd sample of Figure 3 while SD3-28 steps have much smoother texture. The quality degradation is still a concern.
3. There's missed reference in Table 2 for StyleGAN2 + Smart.
4.  How do we find the t* in 5.1? The ablation study is not presented.

**Questions:**

Please refer to weakness.

---

> ### Author Response · Authors · 2024-11-20
>
> Thank you for your constructive feedback of flow-generator matching. Below, we address each of your concerns in detail. Before that, we will first provide a summary of the key contributions of our work.
>
> In this paper, we introduce **Flow Generator Matching (FGM)**, an innovative approach designed to accelerate the sampling of flow matching models into a **one-step generation model**. Our results on CIFAR10 unconditional set a new record **FID score of 3.08** among all flow-matching models. Futhermore, our distillation results on **SD3-medium** demonstrates outstanding performance among other few-step distillation models.  In addition to experimental performance, we propose the **flow-projection identity** to bypass the intractable flow-matching objective, leading to our **practical FGM training objective with theoretial guarantees**—providing a solid foundation for potential advancements in future research.
>
> **Q1: The online flow model**
>
> **A1:** The online flow model $v_\psi$ does represent the implicit vector field, which is used to approximate the vector field of the generator. Since we cannot explicitly compute the generator's vector field (as it is no longer a flow matching model), we learn the flow vector field $v_{\theta,t}$ of the generator distribution through this online flow model $v_\psi$.
>
> On the other hand, the online flow model $v_\psi$ replaces the term $v_{sg[\theta],t}$ in the loss function. Therefore, even though the online flow model cannot be differentiated with respect to $\theta$, it does not affect the gradient's ability to propagate back to the generator through $x_t(\theta)$.  So the equivalent notation for equations (4.11) and (4.12) is presented as follows:
>
> $$
> \mathcal{L}\_{1}(\theta) = \mathbb{E}\_{t, z\sim p_z,  x_0=g_\theta( z), \atop  x_t\sim q_t( x_t| x_0)}
> \{ \\| u_t( x_t) -  v_{\psi}( x_t, t)\\|_2^2 \}
> $$
>
> $$
> \mathcal{L}\_2(\theta) = \mathbb{E}\_{t, z\sim p_z,  x_0=g_\theta( z), \atop  x_t| x_0\sim q_t( x_t| x_0)} [2  (u_t( x_t) -  v_{\psi}( x_t, t) )^T (v_{\psi}(x_t, t) -  u_t( x_t|x_0))]
> $$
>
> **Q2: Different notation for $u_t$**
>
> **A2:** In response to the different notations regarding $u_t(x_t|x_0)$ and $u_t(x_t)$ in above Eq(4.12), it is important to clarify that the first $u_t(x_t)$ represents the pre-trained flow-matching model, which should be considered as a marginal vector field, while the latter term is introduced by **Theorem 4.1 (Flow Product Identity)**; it denotes the conditional vector field used in training of the flow matching model, which is usually the difference between data and sampled Gaussian noise in practice.
>
> **Q3: Abuse of Notation \& Missed Reference**
>
> **A3:** Thank you for pointing out the notation issue on line 203 and missed reference in our table. We have made these adjustments in the revised manuscript to ensure clarity and correctness.
>
> **Q4: Quality Degradation**
>
> **A4:** We present our additional training results in our revised submission **appendix page 24, Figure 8**, comparing the previous results with those from the model trained for more steps. Given that we are fine-tuning a transformer model with 2B parameters, this artifact is typically observed in the early stages of training. This suggests that it can be substantially mitigated, and the overall image quality can also be enhanced with more extensive training.
>
> **Q5: Ablation Study on $t^{*}$**
>
> **A5:** We choose several $t^*=[0.00, 0.25, 0.50, 0.75, 1.00]$ to train from scratch on 512-px, and the qualitative results are presented in our revised submission **appendix page 22, Figure 6**. . Notes that our model parameterization for the ablation can be simplified as
>
> $$
> \hat {x_0} =  z- v_\theta(z, t^*), z\sim \mathcal{N} (\mathbf 0, \mathbf{I})
> $$
>
> The visual results indicate that a suitable range for $t^*$ should be $[0.75, 1.00]$. However, the cost of further determining the optimal choice for $t^*$ is likely to be high and may not yield significant value. A key observation is that as $t^*$ decreases, the structural integrity of the images tends to deteriorate. This phenomenon can be attributed to the property of pre-trained flow matching model. When noise intensity is high, the model primarily focuses on generating the overarching structure of the image. Conversely, at lower noise intensity, the model leans toward creating finer details based on the pre-existing structure. However, in our one-step model, this foundational structure is absent, resulting in divergence.
>
> **We hope our rebuttal have resolved all your concerns. If you still have any concerns, please let us know. We are glad to provide further clarifications as well as more additional experiments.**

---

> ### Author Response · Authors · 2024-11-28
>
> We sincerely appreciate your valuable feedback. We are eager to address any remaining concerns and can provide additional classifications or experiments as needed. A prompt response would greatly assist us in making timely revisions.

---

> > ### Comment · Reviewer_BVRU · 2024-12-03
> >
> > Thank the reviewer for the response. Given such observable flaws in presentation/clarity from the original submission, it would be better to have the paper go through another review round for self consistency and make sure it is understandable by others. I'll remain my rating.

---

> > > ### Author Response · Authors · 2024-12-03
> > >
> > > Thank you for your feedback. We have made significant revisions to improve the presentation and clarity in response to reviewers' feedbacks during rebuttal. We believe the updated version might addresses your concerns effectively. Below is a summary of the major changes made during the revision process:
> > >
> > > 1. As **Reviewer sdTK** and **Reviewer jkBP** wish, **we polish the presentation of the method**. We add a paragraph in Section 4.2 to clarify the difference between FGM and diffusion distillation by pointing out technical challenges. We also add a paragraph in Section 5.1 to compare the training efficiency of FGM and CFM, showing that FGM can significantly surpass CFM on both generation results and training efficiency.
> > > 2. In response to **Reviewer 4qEE** 's suggestion. In Section 4.2, **we have included a comprehensive explanation of the stop $\theta$-gradient technique** utilized within our model. This addition aims to clarify its implementation details and the impact on gradient backpropagation
> > > 3. As suggested by **Reviewer BVRU**, **we have revised the notation in Algorithm 1** for improved clarity. Specifically, we replaced $v_{\text{sg}[\theta]}$ with the actual online flow model $v_\psi$ that we employ. Furthermore, we have provided additional explanations concerning the online flow model $v_\psi$ in Section 4.2 to enhance understanding.
> > > 4. Addressing the feedback from **Reviewer 4qEE** and **Reviewer BVRU**. **we have added detailed pointers in Section 5** that direct readers to the appendix for comprehensive training details, and **expanded our ablation studies**, which can be found in Appendix C. These studies focus on two primary areas: (C.1) generator initialization, and (C.2) the impact of including or excluding regression loss during training.
> > >
> > > **If there are any further issues or concerns, we would be happy to address them promptly.**

---

### Official Review · Reviewer_4qEE · 2024-11-03

**Soundness:** 3
**Presentation:** 2
**Contribution:** 3
**Rating:** 6
**Confidence:** 4

**Summary:**

This paper introduces Flow Generator Matching, an innovative distillation framework for pre-trained flow-matching models. The framework is designed to accelerate sampling by constructing a one-step generator. To achieve this, the authors define a noise-to-image mapping $g_\theta(z)$, and optimize the distance between the vector field of the pre-trained flow model, and the vector field implicitly derived from the one-step generated images and the online flow model. By leveraging a theoretical approximation of this optimization using the flow product identity, the authors develop a fast and efficient one-step flow generator.

**Strengths:**

- Theoretical justification: The proposed distillation formulation is both intuitive and novel in the context of flow matching. Although optimizing $L_{FM}$ directly with respect to $\theta$ is non-trivial, the authors introduce an innovative alternative whose gradient remains aligned with the original objective.

- Specialized distillation methods for flow matching: Given the widespread use of flow matching in various state-of-the-art large-scale generative models (e.g., Movie Gen), effective sampling acceleration is critical. This paper offers a promising approach to address this important contemporary challenge.

- Experiments: Comprehensive empirical evaluations, including unconditional, conditional, and text-to-image experiments, support the effectiveness of the method. The approach demonstrates competitive performance with other flow distillation techniques, achieving lower NFE.

**Weaknesses:**

- Writing & Notations: The connection between the derivations and practical implementations is difficult to follow. For instance, while FGM relies on the online flow model parameterized by $\psi$, as outlined in Algorithm 1, $\psi$ is notably absent in Section 4, despite being a crucial component. Additionally, there is no explanation or justification for the phrase "stop the $\theta$-gradient" mentioned in Line 250, which is a critical implementation detail. From my understanding, FGM is conceptually similar to distribution matching distillation, where both the generator $\theta$ and an online critic $\psi$ are updated alternately. However, this understanding comes from Algorithm 1 rather than the main text. The authors should provide a clear and theoretically grounded explanation of "stop gradient" and revise the derivations to explicitly incorporate the online flow model $\psi$. Moreover, some important information are provided in appendix without pointers, e.g.   model parameterization in L952. Detailed pointers in the main paper would be appreciated.

- Novelty: As previously mentioned, FGM shares significant technical similarities with existing diffusion distillation methods. This overlap may diminish the novelty of the approach and, consequently, inherit some of the limitations of these existing methods, such as reliance on a potentially sub-optimal online flow model.

- Qualitative Results: Although FGM demonstrates better quantitative performance compared to other flow distillation methods, the generated images appear overly "synthetic" and color-saturated (e.g., the rightmost images in Lines 54 and 488), while i admit that this may vary based on human perception. This could be potentially attributed to the image-data-free property proposed as limitations by the authors.

**Questions:**

Overall, the paper is well-structured and presents promising results. However, the derivations and notations could be refined to enhance clarity and facilitate faster understanding. Below are some questions for consideration:

- Ablation Study on Generator Initialization: The generator is initialized with pre-trained flow models, which may be crucial for warm-up and accelerating convergence, especially given that FGM does not utilize real-image datasets. This initialization might also mitigate the OOD gap for pre-trained flow models trained on real images. Conducting an ablation study or providing further analysis could improve understanding and highlight the significance of this choice.

- Bypassing Equation 4.11: Although Equations 4.11 and 4.12 both originate from the FGM loss, the authors opt not to use Equation 4.11, as mentioned in Line 368. This decision represents a significant bypass. Additional analysis, discussion, and empirical evidence are needed to justify this choice. Could the authors elaborate on why Equation 4.11 is deemed unnecessary in practice?

---

> ### Author Response · Authors · 2024-11-20
>
> Thank you for your constructive feedback on our manuscript. We will address these questions below. Before that, we will first provide a summary of the key contributions of our work.
>
> In this paper, we introduce **Flow Generator Matching (FGM)**, an innovative approach designed to accelerate the sampling of flow matching models into a **one-step generation model**. Our results on CIFAR10 unconditional set a new record **FID score of 3.08** among all flow-matching models. Futhermore, our distillation results on **SD3-medium** demonstrates outstanding performance among other few-step distillation models.  In addition to experimental performance, we propose the **flow-projection identity** to bypass the intractable flow-matching objective, leading to our **practical FGM training objective with theoretial guarantees**, which offers a foundation for potential developments in future research.
>
> **Q1: Ablation Study on Generator Initialization**
>
> **A1:** We choose several $t^*=[0.00, 0.25, 0.50, 0.75, 1.00]$ to train from scratch on 512-px, and the qualitative results are presented in our revised submission **appendix page 22, Figure 6**. Notes that our model parameterization for the ablation can be simplified as
> $$
> \hat {x_0} =  z- v_\theta(z, t^*), \quad z\sim \mathcal{N} (\mathbf 0, \mathbf{I})
> $$
>
> The visual results indicate that a suitable range for $t^*$ should be $[0.75, 1.00]$. However, the cost of further determining the optimal choice for $t^*$ is likely to be high and may not yield significant value. A key observation is that as $t^*$ decreases, the structural integrity of the images tends to deteriorate. This phenomenon can be attributed to the property of pre-trained flow matching model. When noise intensity is high, the model primarily focuses on generating the overarching structure of the image. Conversely, at lower noise intensity, the model leans toward creating finer details based on the pre-existing structure. However, in our one-step model, this foundational structure is absent, resulting in divergence.
>
> **Q2: Why regression loss $\mathcal{L}_1$ is unnecessary in practice?**
>
> **A2:** We conduct two experiments on an early checkpoints, one training with both loss $\mathcal{L}_1+\mathcal{L}_2$, another training with only $\mathcal{L}_2$, please check our revised submission **appendix page 23, Figure 7** ,our results show that simply apply the extra regression loss $\mathcal{L}_1$ quickly degrade the performance. From the visual results we can tell that the model trained with $\mathcal{L}_1$ resulting noisy images and quickly corrupted. So the regression term is omitted in our training.
>
> **We hope our rebuttal have resolved all your concerns. If you still have any concerns, please let us know. We are glad to provide further clarifications as well as more additional experiments.**

---

> > ### Comment · Reviewer_4qEE · 2024-11-27
> > **Official Comment by Reviewer 4qEE**
> >
> > Thank you for the clarifications. They address several of my concerns, particularly the explanation of the stop-gradient component. I remain inclined toward acceptance, as the derivation from the intractable flow-matching objective is non-trivial and could inspire future research. However, some unresolved concerns remain as follow.
> >
> > - **Differences from diffusion distillation**: As Reviewer sdTk noted, flow models subsume diffusion models as specific cases. In this context, I find the implications of Lines 313–317 unclear:
> >
> > > (...) flow matching does not imply explicit modeling of either the probability density as the diffusion models do. Therefore, the definitions of distribution divergences cannot be applied to flow models (...)
> >
> > Flow-based generative models can explicitly recover the ODE structure in a form of denoisers (Sec. 2.1, [1]) and generalize probability path definitions, as demonstrated in [2]. Since this 'Differences from diffusion distillation' section is critical to the novelty, more detailed clarification is necessary. While bypassing explicit probability divergence is theoretically meaningful, the novelty could be diminished if the resulting algorithms closely resemble existing methods.
> >
> > - **C.2 Training with regression loss**: While I appreciate the comparative results, a more detailed intuitive explanation, reasoning, or analysis would strengthen this section. Could you elaborate on why omitting Eq. (4.11) improves the results?
> >
> > ---
> > **References**
> >
> > [1] Kim, Beomsu, et al. "Simple ReFlow: Improved Techniques for Fast Flow Models." arXiv preprint arXiv:2410.07815 (2024).
> >
> > [2] Tong, Alexander, et al. "Improving and generalizing flow-based generative models with minibatch optimal transport." arXiv preprint arXiv:2302.00482 (2023).

---

### Official Review · Reviewer_sdTk · 2024-11-03

**Soundness:** 2
**Presentation:** 3
**Contribution:** 2
**Rating:** 3
**Confidence:** 4

**Summary:**

This paper proposes extending score implicit matching from diffusion to flow matching, which is called FGM. The method seems to work well on text-to-image and unconditional generation.

**Strengths:**

1. The paper is well-written and easy to follow.
2. The paper extends the score implicit matching to a flow matching generator.

**Weaknesses:**

1. The novelty of the paper is limited. Since flow matching can be considered a special form of diffusion model, score-based diffusion and flow matching are equivalent. Therefore, rawly extending the implicit score matching for the flow matching model should work well.
2. The evaluation for the text-to-image section should include FID, Recall, ClipScore, Image Reward, PickScore, and AES score. These metrics are often used as standard metrics to evaluate generative models.
3. Since this method is closely related to implicit score matching, the author should include a background section for implicit score matching.

**Questions:**

My biggest concern is the novelty and originality of this paper's idea. There is a list of works working on score implicit models, such as [1,2,3]. Extending the framework from diffusion to flow matching is not interesting and does not introduce something new.

[1]: Diff-Instruct: A Universal Approach for Transferring Knowledge From Pre-trained Diffusion Models

[2]: One-Step Diffusion Distillation through Score Implicit Matching

[3]: Score identity Distillation: Exponentially Fast Distillation of Pretrained Diffusion Models for One-Step Generation

---

> ### Author Response · Authors · 2024-11-20
>
> We sincerely appreciate your constructive criticism and suggestions. In response to your concerns, we would like to clarify the following points. Before that, we will first provide a summary of the key contributions of our work.
>
> In this paper, we present **Flow Generator Matching (FGM)**, an innovative approach amied at accelerating the sampling of flow matching models into a **one-step generation model**. Our results on CIFAR10 unconditional achieve a remarkable **FID score of 3.08** among all flow-matching models. Futhermore, our distillation results on **SD3-medium** demonstrates outstanding performance among other few-step distillation models.  In addition to experimental performance, we propose the **flow-projection identity** to bypass the intractable flow-matching objective, leading to our **practical FGM training objective with theoretial guarantees**, which lays the groundwork for potential advancements in future research. In contrast to earlier score-based implicit distillation methods, our approach is distinct in its emphasis on the flow-matching objective rather than relying on inapplicable score functions in flow matching.
>
> **Q1: Novelty and originality of this paper's idea**
>
> **A1:** Previous implicit models have largely relied on diffusion models. A key distinction between flow-matching and diffusion models is that flow matching does not inherently imply score functions, rendering the definitions of distribution divergences inapplicable.
>
> Our contribution resolves this issue by **focusing on the flow matching objective**, rather than on distribution divergence.  To the best of our knowledge, the flow-generator-matching is one of the few attempts to introduce one-step distillation method based on the flow matching model without explicitly involving probability divergences. Interestingly, FGM bypass the need for probability divergence through directly handling the flow-matching objective. To do so, we introduce a novel flow-projection identity and deriving an equivalent loss that minimizes the intractable flow-matching objective. However, we do appreciate previous works, especially the score-identity distillation and the score-implicit matching. Both works focuses on diffusion models which have an straightforward probability interpretation rather than flow-matching models.
>
> **Q2: Extra Evaluation**
>
> **A2:** We have followed your suggestion to conduct further evaluation on the COCO 2017 validation set with 5,000 samples. This more comprehensive evaluation demonstrates that our one-step distillation model closely matches the performance of the teacher model in many aspects and even surpasses the results of multi-step distillation on several metrics.
>
> | Model        | Steps | FID       | CLIP Score | Image Reward | Aesthetic Score | Pick Score |
> | ------------ | ----- | --------- | ---------- | ------------ | --------------- | ---------- |
> | SD3(Teacher) | 28    | 23.15     | 32.10      | 0.92         | 5.27            | 0.223      |
> | Hyper-SD3    | 4     | 71.64     | 31.57      | **0.83**     | 5.30            | **0.225**  |
> | Flash-SD3    | 4     | 70.67     | 31.91      | 0.65         | 5.38            | 0.220      |
> | Ours         | **1** | **32.75** | **31.92**  | 0.78         | **5.39**        | 0.221      |
>
> **We hope our rebuttal have resolved all your concerns. If you still have any concerns, please let us know. We are glad to provide further clarifications as well as more additional experiments.**

---

> ### Author Response · Authors · 2024-11-28
>
> We sincerely appreciate your valuable feedback. We are eager to address any remaining concerns and can provide additional classifications or experiments as needed. A prompt response would greatly assist us in making timely improvements.

---

> ### Author Response · Authors · 2024-12-03
>
> As we approach the conclusion of our rebuttal, we would like to kindly **summarize our key points** and **express our eagerness for your final feedback.** We have made significant revisions to enhance the manuscript in response to the reviewers' feedback. Below is a summary of the major changes implemented during the revision process:
>
> 1. As **Reviewer sdTK** and **Reviewer jkBP** wish, **we polish the presentation of the method**. We add a paragraph in Section 4.2 to clarify the difference between FGM and diffusion distillation by pointing out technical challenges. We also add a paragraph in Section 5.1 to compare the training efficiency of FGM and CFM, showing that FGM can significantly surpass CFM on both generation results and training efficiency.
> 2. In response to **Reviewer 4qEE** 's suggestion. In Section 4.2, **we have included a comprehensive explanation of the stop $\theta$-gradient technique** utilized within our model. This addition aims to clarify its implementation details and the impact on gradient backpropagation
> 3. As suggested by **Reviewer BVRU**, **we have revised the notation in Algorithm 1** for improved clarity. Specifically, we replaced $v_{\text{sg}[\theta]}$ with the actual online flow model $v_\psi$ that we employ. Furthermore, we have provided additional explanations concerning the online flow model $v_\psi$ in Section 4.2 to enhance understanding.
> 4. Addressing the feedback from **Reviewer 4qEE** and **Reviewer BVRU**. **we have added detailed pointers in Section 5** that direct readers to the appendix for comprehensive training details, and **expanded our ablation studies**, which can be found in Appendix C. These studies focus on two primary areas: (C.1) generator initialization, and (C.2) the impact of including or excluding regression loss during training.
>
> **If you have any additional issues or concerns, we would be happy to address them promptly.**

---

### Official Review · Reviewer_jkBP · 2024-11-04

**Soundness:** 3
**Presentation:** 4
**Contribution:** 3
**Rating:** 6
**Confidence:** 3

**Summary:**

The paper introduces the **Flow Generator Matching (FGM) objective** for distilling a one-step generator from pretrained flow-matching models. It provides theoretical guarantees that the proposed objective yields the same gradient, with respect to the parameters of the one-step generator, as the standard flow-matching loss. The paper demonstrates competitive and, in some cases, superior results in both unconditional generation on CIFAR-10 and large-scale text-to-image generation, achieved by distilling Stable Diffusion 3 (SD3).

**Strengths:**

1. The paper is well-written and easy to follow.
2. The proposed algorithm is elegantly designed, alternating the optimization of the parameters of the one-step generator, $\theta$, and an auxiliary flow model, $\psi$. The correctness and effectiveness of the algorithm are supported by Theorems 4.1 and 4.2.
3. The paper conducts thorough empirical evaluations of the proposed method across various tasks, including unconditional generation on CIFAR-10 and large-scale text-to-image generation, demonstrating superior results.

**Weaknesses:**

1. The proposed method requires training an auxiliary flow model with parameters $\psi$, designed to generate the implicit flow determined by the one-step generator. Is this auxiliary model used after training?
2. The paper does not include a discussion of training efficiency or speed. One concern is that the proposed method might be slower compared to distillation approaches like Consistency Flow Matching (Yang, Ling, et al., "Consistency Flow Matching: Defining Straight Flows with Velocity Consistency"). Could you compare or discuss the training efficiency of the proposed methods with other common baselines?

**Questions:**

Please see weaknesses part.

---

> ### Author Response · Authors · 2024-11-20
>
> We appreciate the your feedback and would like to address the following points. Before that, we will first provide a summary of the key contributions of our work.
>
> In this paper, we introduce **Flow Generator Matching (FGM)**, an innovative approach designed to accelerate the sampling of flow matching models into a **one-step generation model**. Our results on CIFAR10 unconditional set a new record **FID score of 3.08** among all flow-matching models. Futhermore, our distillation results on **SD3-medium** demonstrates outstanding performance among other few-step distillation models.  In addition to experimental performance, we propose the **flow-projection identity** to bypass the intractable flow-matching objective, leading to our **practical FGM training objective with theoretial guarantees**, thus laying a solid groundwork for future research advancements.
>
> **Q1: Is this auxiliary model used after training?**
>
> **A1:** The auxiliary model would be **no longer needed** when the training is finished. Once our one-step generator has almost converged, the auxiliary model should be similar to the per-trained flow-matching model. In practice, we only use keep one-step generator for text-to-image generations. However, we believe that the study on how to further use knowledge remained in the auxiliary model is an interesting research topic in future.
>
> **Q2: Training efficiency compared with other common baselines**
>
> **A2:** On distillation of Stable-diffusion-3-medium, a 2B transformer model, our **full parameter** training spent approximate 25 H800 days to achieve the reported results with 23k optimization step. It is important to note that this duration includes a significant amount of exploratory testing. Once all hyperparameters are determined, the actual training time required to develop a production-ready model may be considerably shorter.
>
> As for other common baselines, Flash-SD3 was trained for approximately 50 hours on 2 H100 GPUs utilizing LoRA, with **only 90.4 million trainable parameters**. The training duration for Hyper-SD3 is currently unknown. Although our model requires a significantly longer training time compared to Flash-SD3, the adoption of LoRA may substantially reduce the training costs, an avenue we plan to explore further.
>
> **We hope our rebuttal have resolved all your concerns. If you still have any concerns, please let us know. We are glad to provide further clarifications as well as more additional experiments.**

---

> ### Author Response · Authors · 2024-11-28
>
> We sincerely appreciate your valuable feedback. We are eager to address any remaining concerns and can provide additional classifications or experiments as needed. A prompt response would greatly assist us in making timely improvements.

---

> ### Author Response · Authors · 2024-12-03
>
> As we approach the conclusion of our rebuttal, we would like to kindly **summarize our key points** and **express our eagerness for your final feedback.** We have made significant revisions to enhance the manuscript in response to the reviewers' feedback. Below is a summary of the major changes implemented during the revision process:
> 1. As **Reviewer sdTK** and **Reviewer jkBP** wish, **we polish the presentation of the method**. We add a paragraph in Section 4.2 to clarify the difference between FGM and diffusion distillation by pointing out technical challenges. We also add a paragraph in Section 5.1 to compare the training efficiency of FGM and CFM, showing that FGM can significantly surpass CFM on both generation results and training efficiency.
> 2. In response to **Reviewer 4qEE** 's suggestion. In Section 4.2, **we have included a comprehensive explanation of the stop $\theta$-gradient technique** utilized within our model. This addition aims to clarify its implementation details and the impact on gradient backpropagation
> 3. As suggested by **Reviewer BVRU**, **we have revised the notation in Algorithm 1** for improved clarity. Specifically, we replaced $v_{\text{sg}[\theta]}$ with the actual online flow model $v_\psi$ that we employ. Furthermore, we have provided additional explanations concerning the online flow model $v_\psi$ in Section 4.2 to enhance understanding.
> 4. Addressing the feedback from **Reviewer 4qEE** and **Reviewer BVRU**. **we have added detailed pointers in Section 5** that direct readers to the appendix for comprehensive training details, and **expanded our ablation studies**, which can be found in Appendix C. These studies focus on two primary areas: (C.1) generator initialization, and (C.2) the impact of including or excluding regression loss during training.
>
> **If you have any additional issues or concerns, we would be happy to address them promptly.**

---

### Author Response · Authors · 2024-11-24
**Paper Revision**

# Dear Reviewers and Area Chair,

We thank all reviewers for their useful feedback. We have made significant updates to our draft to improve the overall writing quality, provide more clarifications, and add additional experimental results. We have highlighted the changes in the draft in blue color. Here are the major changes:

1. As **Reviewer sdTK** and **Reviewer jkBP** wish, **we polish the presentation of the method**. We add a paragraph in Section 4.2 to clarify the difference between FGM and diffusion distillation by pointing out technical challenges. We also add a paragraph in Section 5.1 to compare the training efficiency of FGM and CFM, showing that FGM can significantly surpass CFM on both generation results and training efficiency.

2. In response to **Reviewer 4qEE** 's suggestion. In Section 4.2, **we have included a comprehensive explanation of the stop $\theta$-gradient technique** utilized within our model. This addition aims to clarify its implementation details and the impact on gradient backpropagation

3. As suggested by **Reviewer BVRU**, **we have revised the notation in Algorithm 1** for improved clarity. Specifically, we replaced $v_{\text{sg}[\theta]}$  with the actual online flow model $v_\psi$ that we employ. Furthermore, we have provided additional explanations concerning the online flow model $v_\psi$ in Section 4.2 to enhance understanding.

4. Addressing the feedback from **Reviewer 4qEE** and **Reviewer BVRU**. **we have added detailed pointers in Section 5** that direct readers to the appendix for comprehensive training details, and **expanded our ablation studies**, which can be found in Appendix C. These studies focus on two primary areas: (C.1) generator initialization, and (C.2) the impact of including or excluding regression loss during training.

**Change 1** improved the overall presentation of the paper a lot.  **Change 2** aided readers in comprehensively understanding the purpose and functionality of the stop-gradient technique within our model. **Change 3** clarified the notation utilized in Algorithm 1, rendering it more accessible and easier for readers to follow. **Change 4** expanded our ablation studies, which substantially enrich our findings and provide deeper insights into the research.

We appreciate constructive suggestions from all reviewers that help strengthen our draft.

Best,

Authors of submission #8666

---

### Meta-Review · Area_Chair_Vcov · 2024-12-19

**Metareview:**

This paper suggest a distillation method based on the Flow Matching loss where the idea is to optimize a “generator” $g_\theta$ that maps noise to sample in one step by defining the student data distribution $p_{\theta,0}$ via $g_\theta$ and minimize the FM loss to match it to the teacher data distribution defined by the given pre-trained model.

Concerns about this paper in its current state include its somewhat limited relative added contribution compared to recent implicit score 1-step distillation methods that take a similar approach but for score parameterization, the quality of the results presented, and lacking/unclear presentation. We still feel the paper is of value as a promising flow distillation method and encourage the authors to revise their paper and resubmit in the future.

**Additional Comments On Reviewer Discussion:**

No additional comments.

---

### Decision · Program_Chairs · 2025-01-22

Reject